# Fight Fire with Fire: Multi-biased Interactions in Hard-Thresholding

## Abstract

$\ell_0$ constrained optimization is widely used in machine learning, especially for high-dimensional problems, as it effectively promotes sparse learning. A prominent technique for solving these problems is hard-thresholding gradient descent. However, the inherent expansibility of hard-thresholding operators can lead to convergence issues, necessitating strategies to accelerate the algorithm. In this article, we believe the random hard-thresholding algorithm can be interpreted as an equivalent biased gradient algorithm. By introducing appropriate biases, we can mitigate some of the issues of hard-thresholding and enhance convergence. We categorize the biases into memory-biased and recursive-biased, examining their distinct applications within hard-thresholding algorithms. Next, we explore the zeroth-order versions of these algorithms, which introduce additional biases from zeroth-order gradients. Our findings indicate that recursively bias effectively counteracts some of the issues caused by hard-thresholding, resulting in improved performance for first-order algorithms. Conversely, due to the accumulation of errors from zeroth-order gradients during recursive bias, the performance of zeroth-order algorithms is inferior to that influenced by historical gradients. To address these insights, we propose the SARAHT and BVR-SZHT algorithms for first-order and zeroth-order hard-thresholding, respectively, both of which demonstrate faster convergence speeds compared to previous methods. We validate our hypotheses through black-box adversarial experiments and ridge regression evaluations.

## 1 Introduction

$\ell_0$ constrained optimization is a fundamental technique in large-scale machine learning, especially in high-dimensional settings where sparsity is crucial (Fan & Li, 2001; Zhang, 2010). It promotes sparse learning, offering benefits like reduced memory usage, lower computational costs, and improved efficiency. In this study, we address the following problem:

$$\min_{x \in \mathbb{R}^d} f(x) = \frac{1}{n} \sum_{i=1}^{n} f_i(x), \quad \text{s.t.} \|x\|_0 \le k,$$

where $f(x)$ represents the empirical risk, and $\|x\|_0$ denotes the number of non-zero elements. The $\ell_0$ constraint makes this problem NP-hard, limiting the use of traditional methods. Unlike $\ell_1$ optimization (e.g., LASSO), $l_0$ optimization naturally has lower computational costs, making $l_0$-based algorithms faster in general. Additionally, in scenarios requiring strict sparsity, $\ell_1$ often struggles because it is difficult to directly specify the sparsity level.

To solve this problem, we are particularly interested in gradient hard-thresholding methods (Raskutti et al., 2011; Jain et al., 2014; Nguyen et al., 2017; Yuan et al., 2017), which are used for obtaining approximate solutions to $\ell_0$ constrained optimization problems. This technique alternates between a gradient step and the application of the hard threshold operator $\mathcal{H}_k(x)$, which retains the top $k$ elements of $x$ while setting all other directions to zero. The gradient hard-thresholding iteration is given by:

$$x^{t+1} = \mathcal{H}_k(x^t - \eta g(x^t)), \tag{1}$$

where $g(x^t)$ is the gradient oracle.

Hard-thresholding was first used for its full gradient form (Jain et al., 2014). (Nguyen et al., 2017) developed a stochastic gradient descent Stochastic Gradient Descent(SGD) version of hard-thresholding known as StoIHT. Nevertheless, StoIHT's convergence condition is overly stringent for practical applications (Li et al., 2016). To address this issue, (Zhou et al., 2018), (Shen & Li, 2017) and (Li et al., 2016) implemented variance reduction techniques to improve the performance of StoIHT in real-world problem-solving. Furthermore, (de Vazelhes et al., 2022) designed the stochastic zeroth-order hard-thresholding algorithm and found that the expansion of hard-thresholding gradients and the errors in zeroth-order gradients can create a kind of antagonism, causing the algorithm to struggle with convergence. (Yuan et al., 2024) found that reducing the variance could help mitigate this conflict.

In previous works, the gradient oracle process and the hard-thresholding iterative process were treated separately, without examining their interrelationship in influencing algorithm convergence. In this paper, We view the stochastic gradient decent step and the hard-thresholding step as a whole and consider them as an equivalent gradient $\nabla_{HT}^t = (x^t - x^{t+1})/\eta$. This approach enables us to reinterpret the hard-thresholding algorithm as a specific type of biased gradient algorithm. By doing so, we uncover the potential to enhance convergence by designing appropriate biased gradient oracles.

Recently, there has been increasing interest in SGD using biased gradient oracles, which has been explored in various studies across multiple domains. A notable example includes zeroth-order methods, such as in optimizing black-box functions (Nesterov & Spokoiny, 2017) or in generating adversarial examples in deep learning (Moosavi-Dezfooli et al., 2017; Chen et al., 2017). Many zeroth-order training techniques leverage biased gradient oracles (Liu et al., 2018; Bergou et al., 2020), and biased estimators can outperform their unbiased counterparts in specific contexts (Beznosikov et al., 2020).

Actually, there has been a recent surge of interest in SGD with biased gradient oracles, which has been studied in several papers and applied in different domains. A typical example is zeroth-order methods, which are often utilized when there is no access to unbiased gradients, e.g., for optimization of black-box functions (Nesterov & Spokoiny, 2017) or for finding adversarial examples in deep learning (Moosavi-Dezfooli et al., 2017; Chen et al., 2017). Many zeroth-order training methods exploit biased gradient oracles (Liu et al., 2018; Bergou et al., 2020). Moreover, biased estimators may show better performance over their unbiased equivalents in certain settings (Beznosikov et al., 2020). This raises some interesting questions:

---

In algorithms that utilize multiple biased gradient oracles, how do these biases interact? More specifically, how do they affect the hard-thresholding algorithm when viewed as an equivalent biased algorithm?

---

In this paper, we investigate how appropriate biases can mitigate the challenges posed by hard-thresholding and enhance convergence. We categorize these biases into memory-biased and recursive-biased, examining their distinct applications within hard-thresholding algorithms. Additionally, we explore the zeroth-order versions of these algorithms, which introduce further biases from zeroth-order gradients. Our findings indicate that recursively bias effectively counteracts some issues caused by hard-thresholding, leading to improved performance in first-order algorithms. However, the accumulation of errors from zeroth-order gradients during recursively bias results in inferior performance compared to historical gradients. To address these insights, we propose the SARAH-HT and BVR-SZHT algorithms for first-order and zeroth-order hard-thresholding, respectively, both demonstrating faster convergence speeds compared to previous methods. We validate our hypotheses through black-box adversarial experiments and ridge regression evaluations, providing a thorough examination of the effects of multiple biases on convergence and their integration with the hard-thresholding operator.

1. To the best of our knowledge, this is the first time a biased gradient approach is used to analyze the hard-thresholding algorithm, accelerating the algorithm through a biased gradient oracle.

2. We analyze the relationships between multiple biases and zeroth-order bias, providing a method to potentially eliminate these biases.

3. We propose a series of (zeroth-order) hard-thresholding bias algorithms and analyze their convergence, showing improved convergence speed compared to existing algorithms.

## 2 UNDERSTANDING BIAS IN OPTIMIZATION

In this section, we will provide a detailed introduction to several forms of bias mentioned in this paper and explain how these biases affect convergence. We use $\|x\|$ to denote the Euclidean norm for a vector, $\|x\|_\infty$ to denote the maximum absolute component of that vector, and $\|x\|_0$ to denote the $\ell_0$ norm (which is not a proper norm).

### 2.1 BIAS AND CONVERGENCE

It is well known that the mean squared error (MSE) of gradient estimation $\mathbb{E}\|g(x) - \nabla f(x)\|^2$ is a key factor in evaluating the quality of the gradient oracle $g(x)$. A smaller MSE usually indicates a faster convergence rate. In fact:

$$\mathbb{E}\|g(x) - \nabla f(x)\|^2 = \|\mathbb{E}g(x) - \nabla f(x)\|^2 + \mathbb{E}\|g(x) - \mathbb{E}[g(x)]\|^2, \tag{2}$$

where $\mathbb{E}\|g(x) - \mathbb{E}[g(x)]\|^2$ is the variance of $g(x)$ and $\|\mathbb{E}g(x) - \nabla f(x)\|^2$ is the squared norm of the bias $g(x)$. This suggests that bias can often lead to non-convergence. However, many algorithms reduce variance through specific biases, thereby decreasing the MSE and accelerating convergence. We refer to this as the biased gradient descent oracle.

**Remark 1** *In hard-thresholding algorithms, the MSE of $g(x)$ does not completely determine the convergence of the algorithm. However, we can use $\nabla_{HT}^t = \frac{x^{t+1} - x^t}{\eta}$ as a substitute.*

### 2.2 BIASED VARIANCE REDUCE ESTIMATION

Biased gradient descent estimation is used in many algorithms, such as BSVRG, BSAGA, and SARAH. Their estimation are:

**B-SAGA:** $g(x^t) \stackrel{\text{def}}{=} \frac{1}{\theta}\left(\nabla f_{j_t}(x_t) - \nabla f_{j_t}(\varphi_t^{j_t})\right) + \frac{1}{n}\sum_{i=1}^n \nabla f_i(\varphi_t^i)$,

**B-SVRG:** $g(x^t) \stackrel{\text{def}}{=} \begin{cases} \frac{1}{n}\sum_{i=1}^n \nabla f_i(\varphi) & \text{for } t \in \nu\mathbb{N}_0 \\ \nabla f_{j_t}(x_t) - \nabla f_{j_t}(x_{t-1}) + \frac{1}{n}\sum_{i=1}^n \nabla f_i(\varphi) & \text{o.w.} \end{cases}$

**SARAH:** $g(x^t) \stackrel{\text{def}}{=} \begin{cases} \frac{1}{n}\sum_{i=1}^n \nabla f_i(\varphi) & \text{for } t \in \nu\mathbb{N}_0 \\ \nabla f_{j_t}(x_t) - \nabla f_{j_t}(x_{t-1}) + g(x^{t-1}) & \text{o.w.} \end{cases}$

Here, $\varphi$ means the historical information. The parameter $\nu$ represents how many steps occur between full gradient evaluations.

These algorithms, through specific configurations, reduce the MSE even in the presence of bias. The vast majority of such algorithms satisfy the following BMSE assumption.

**Assumption 1** *(Driggs et al., 2022) (Bounded MSE) The stochastic gradient estimator $g(x^t)$ is said to satisfy the BMSE $(M_1, M_2, \rho_M, \rho_F, m)$ property with parameters $M_1, M_2 \geq 0, \rho_M, \rho_F \in (0,1]$ and $m \geq 1$ if there exist sequences $\mathcal{M}_t$ and $\mathcal{F}_t$ such that*

$$\sum_{t=ms}^{m(s+1)-1} \mathbb{E}\left[\left\|g(x^t) - \nabla f\left(x^t\right)\right\|^2\right] \leq \mathcal{M}_{ms},$$

*and the following bounds hold:*

$$\mathcal{M}_{ms} \leq (1 - \rho_M)^m \mathcal{M}_{m(s-1)} + \mathcal{F}_{ms} + \frac{M_1}{n}\sum_{t=ms}^{m(s+1)-1}\sum_{i=1}^n \mathbb{E}\left[\left\|\nabla f_i\left(x^{t+1}\right) - \nabla f_i\left(x^t\right)\right\|^2\right];$$

$$\mathcal{F}_{ms} \leq \sum_{\ell=0}^{s} \frac{M_2 \left(1 - \rho_F\right)^{m(s-\ell)}}{n} \sum_{t=ms}^{m(s+1)-1} \sum_{i=1}^{n} \mathbb{E}\left[\left\|\nabla f_i\left(x^{t+1}\right) - \nabla f_i\left(x^t\right)\right\|^2\right].$$

We can broadly categorize these configurations into two parts. That is:

**Definition 1** *(Memory-biased gradient oracle) The stochastic gradient oracle $g(x^t)$ is memory-biased with parameters $\theta > 0, B_1 \geq 0$, and $m \geq 1$ if*

$$\nabla f\left(x^t\right) - \mathbb{E}_k g(x^t) = \left(1 - \frac{1}{\theta}\right)\left(\nabla f\left(x^t\right) - \frac{1}{n}\sum_{i=1}^{n} \nabla f_i\left(\varphi_k^i\right)\right),$$

*for some $\left\{\varphi_k^i\right\}_{i=1}^{n} \subset \{x_\ell\}_{\ell=0}^{t-1}$, and for any $s \in \mathbb{N}_0$,*

$$\sum_{k=ms}^{m(s+1)-1} \frac{1}{n}\sum_{i=1}^{n} \mathbb{E}\left[\left\|x^t - \varphi_k^i\right\|^2\right] \leq B_1 \sum_{k=ms}^{m(s+1)-1} \mathbb{E}\left[\left\|x^t - x^{t-1}\right\|^2\right].$$

*The parameter $\frac{1}{\theta}$ represents the degree of bias. When $\theta = 1$, the algorithm is unbiased.*

**Definition 2** *(Recursive-biased gradient oracle) For any sequence $\{x_k\}$, let $\widetilde{\nabla}_k$ be a stochastic gradient oracle generated from the points $\{x_\ell\}_{\ell=0}^{k}$. This estimator is recursive-biased with $\nu \geq 1$ if*

$$\nabla f\left(x_k\right) - \mathbb{E}_k g(x^t) = \begin{cases} 0 & \text{for } k \in \nu\mathbb{N}_0, \\ \left(\nabla f\left(x_{k-1}\right) - \widetilde{\nabla}_{k-1}\right) & o.w. . \end{cases}$$

*The parameter $\nu$ represents how many steps occur between full gradient evaluations.*

BSVRG and BSAGA have memory-biased gradient oracle and SARAH has recursive-biased gradient oracle(Driggs et al., 2022). Through this classification, we can systematically study the impact of such biases in greater detail in section 3.

## 2.3 HARD-THRESHOLDING OPERATOR

As described in Section 1, we can view the stochastic gradient decent step and the hard-thresholding step as a whole and consider them as an equivalent gradient $\nabla_{HT}^t = (x^t - x^{t+1})/\eta$. Following this reasoning:

**Lemma 1** *For any $\{x^t\}$ that satisfies $x^{t+1} = \mathcal{H}_k(x^t - \eta g(x^t))$ and $x \in \mathbb{R}^d$, we have:*

$$(\gamma_k - \frac{1}{2})\|x^{t+1} - x\|^2 + \frac{1}{2}\|x^t - x\|^2 - \frac{1}{2}\eta^2\|\nabla_{HT}^t\|^2 \geq \eta\left\langle g(x^t), x^{t+1} - x\right\rangle,$$

*where $\gamma_k = \sqrt{k^*/k}/2$ is the hard-thresholding coefficient.*

In addition, we use two assumptions, which are widely adopted in hard-thresholding algorithm (Li et al., 2016; Nguyen et al., 2017).

**Assumption 2 (Restricted strong convexity (RSC) (Li et al., 2016; Nguyen et al., 2017))** *A differentiable function $f$ is restricted $v_s$-strongly convex at sparsity $s$ if there exists a generic constant $v_s > 0$ such that for any $x, x' \in \mathbb{R}^d$ with $\|x - x'\|_0 \leq s$, we have:*

$$f(x) - f(x') - \left\langle \nabla f(x'), x - x'\right\rangle \geq \frac{v_s}{2}\|x - x'\|_2^2. \tag{3}$$

**Assumption 3 (Restricted strong smoothness (RSS) (Li et al., 2016; Nguyen et al., 2017))** *For any $i \in [n]$, a differentiable function $f_i$ is restricted $L_s$-strongly smooth at sparsity level $s$ if there exists a generic constant $L_s > 0$ such that for any $x, x' \in \mathbb{R}^d$ with $\|x - x'\|_0 \leq s$, we have*

$$\|\nabla f_i(x) - \nabla f_i(x')\| \leq L_s\|x - x'\|.$$

We assume that the objective function $f(x)$ satisfies the RSC condition and that each component function $\{f_i(x)\}_{i=1}^n$ satisfies the RSS condition. We also define the restricted condition number as $\kappa_s = v_s/L_s$. This assumption ensures that the objective function behaves like a strongly convex and smooth function over a sparse domain, even when it is non-convex.

## 2.4 ZEROTH-ORDER GRADIENT

The zeroth-order gradient oracle obtained by Gaussian smoothing is a typical scenario of biased gradients (Stich, 2020) In hard-thresholding algorirthm, A commonly used zeroth-order estimation is(de Vazelhes et al., 2022; Yuan et al., 2024)

$$\hat{\nabla} f(x) = \frac{d}{q\mu} \sum_{i=1}^{q} (f(x + \mu \boldsymbol{u}_i) - f(x)) \boldsymbol{u}_i, \tag{4}$$

where each random direction $\boldsymbol{u}_i$ is a unit vector sampled uniformly from the set $\{\boldsymbol{u} \in \mathbb{R}^d : \|\boldsymbol{u}\|_0 \le s_2, \|\boldsymbol{u}\| = 1\}$, $q$ is the number of random unit vectors, and $\mu > 0$ is a constant called the *smoothing radius* (typically taken as small as possible, but no too small to avoid numerical errors). To obtain these vectors, we can first sample a random set of coordinates $S$ of size $s_2$ from $[d]$. Following, we sample a random vector $\boldsymbol{u}$ supported on $S$, in other words, uniformly sampled from the set $\{u \in \mathbb{R}^d : \boldsymbol{u}_{[d]-S} = \boldsymbol{0}, \|\boldsymbol{u}\| = 1\}$.

# 3 MUTI-BIAS INTERACTION IN HARD-THRESHOLDING

In this section, we will examine the performance of memory-biased and recursive-biased in different scenarios to investigate the interaction between biases. Specifically, we will first study the performance of memory-biased gradient and recursive-biased gradient oracle in the first-order hard-thresholding algorithm, and then analyze their performance in the zeroth-order hard-thresholding algorithm.

## 3.1 FIRST-ORDER HARD-THRESHOLDING

In first-order algorithms, from Lemma 1 we have

**Lemma 2** *Suppose $g(x)$ is a recursively biased oracle. Suppose $f$ satisfies the RSC condition with $v_s \ge 0$. The following inequality holds:*

$$\eta \mathbb{E}[f(x^{t+1}) - f(x^*) + (\frac{1}{2} - \gamma_k)\|x^{t+1} - x^*\|^2] \le \mathbb{E}[\frac{1}{2}\|x^t - x^*\|^2 - \frac{\eta^2}{2}\|\nabla_{HT}^t\|^2 \\ + \eta \mathbb{E}\left\langle \nabla_F f\left(x^t\right) - g(x^t), x^t - x^*\right\rangle]. \tag{5}$$

**Remark 2** *In (5), $\|x^t - x^*\|^2$ is the convergence term. $\eta \mathbb{E}\left\langle \nabla_F f\left(x^t\right) - g(x^t), x^t - x^*\right\rangle - \frac{\eta^2}{2}\|\nabla_{HT}^t\|^2$ implies the bias of hard-thresholding. And $\eta \mathbb{E}\left\langle \nabla_F f\left(x^t\right) - g(x^t), x^t - x^*\right\rangle$ is the biased caused by recursive bias, since when the gradient oracle is unbiased, this term is $0$.*

For $\mathbb{E}\left\langle \nabla_F f\left(x^t\right) - g(x^t), x^t - x^*\right\rangle$, when $g(x)$ is recursively biased, we have:

$$\mathbb{E}\left\langle \nabla_F f\left(x^t\right) - g(x^t), x^t - x^*\right\rangle \stackrel{(1)}{=} \mathbb{E}\left\langle \nabla_F f\left(x^t\right) - \mathbb{E}_t g(x^t), x^t - x^*\right\rangle \\ \stackrel{(2)}{\le} \mathbb{E}\left\langle \nabla_F f\left(x^{t-1}\right) - g(x^{t-1}), x^t - x^*\right\rangle.$$

We can pass the conditional expectation $\mathbb{E}_{t-1}$ into the second inner-product in (1) because $x_{t-1}$ is independent of $j_{t-1}$. Inequality (2) uses the definition of a recursively biased gradient oracle. This

is a recursive inequality, and expanding the recursion gives

$$\mathbb{E}\left\langle \nabla_F f\left(x^{t-1}\right) - g(x^{t-1}), x^t - x^* \right\rangle$$

$$= \mathbb{E}\left\langle \nabla_F f\left(x^{t-1}\right) - g(x^{t-1}), x^t - x^{t-1} \right\rangle + \mathbb{E}\left\langle \nabla_F f\left(x^{t-1}\right) - g(x^{t-1}), x^{t-1} - x^* \right\rangle$$

$$\leq \sum_{l=vs+1}^{t-1} \mathbb{E}\left\langle \nabla_F f\left(x^{l-1}\right) - g(x^{l-1}), x^l - x^{l-1} \right\rangle + \mathbb{E}\left\langle \nabla_F f\left(x^{vs}\right) - g(x^{vs}), x^{vs} - x^* \right\rangle$$

$$\overset{1}{=} \sum_{l=vs+1}^{t-1} \mathbb{E}\left\langle \nabla_F f\left(x^{l-1}\right) - g(x^{l-1}), x^l - x^{l-1} \right\rangle,$$

here equation 1 is because $(x^{\nu s} - x^*)_{F^c} = 0$ and $(\nabla_F f\left(x^{vs}\right) - g(x^{vs}))_F = 0$.

**Remark 3** *We should mention that $\nabla_F f\left(x^{vs}\right) \neq g(x^{vs})$. $\left\langle \nabla_F f\left(x^{l-1}\right) - g(x^{l-1}), x^l - x^* \right\rangle$ relies on hard-thresholding operator to make sure $(x^{\nu s} - x^*)_{F^c} = 0$, which means that the bias is partly canceled by hard-thresholding.*

Therefore, we have:

**Lemma 3** *If $g(x^t)$ is recursively biased, for any $\epsilon > 0$, there is*

$$\sum_{t=\nu s+1}^{\nu(s+1)-1} \left\| \mathbb{E}\left\langle \nabla_F f\left(x^{t-1}\right) - g(x^{t-1}), x^t - x^* \right\rangle \right\| \leq \nu \sum_{t=\nu s+1}^{\nu(s+1)-1} \mathbb{E}\left[ \frac{\epsilon}{2} \left\| \nabla_F f\left(x_t\right) - g(x^t) \right\|^2 + \frac{1}{2\epsilon} \left\| x^{t+1} - x^t \right\|^2 \right].$$

(6)

By BMSE condition, we know $\mathbb{E}\left\langle \nabla_F f\left(x^{t-1}\right) - g(x^{t-1}), x^t - x^* \right\rangle$ can be controlled by $\nabla_{HT}$ during the iteration. This implies that the bias caused by the hard threshold in (5) will be partially canceled out.

**Remark 4** *From Lemmas 2 and 3, we conclude that the bias of the recursively biased algorithm is partially canceled when $t = \nu s$. Consequently, during the iterations of this algorithm, the bias of the equivalent gradient is also mitigated. This suggests that the recursively biased algorithm for the hard-threshold can counteract some bias, thus accelerating convergence.*

**Lemma 4** *Suppose $g(x)$ is a first-order memory-biased oracle. Suppose $\theta \geq 1$ and that $f$ satisfies the RSC condition with $v_s \geq 0$. the following inequality holds:*

$$\eta \mathbb{E}[f(x^{t+1}) - f(x^*) + (\frac{1}{2} - \gamma_k)\|x^{t+1} - x^*\|^2] \leq \eta \left\langle g(x^t) - \nabla_F f(x^t), x^t - x^{t+1} \right\rangle + \frac{1}{2}\|x^t - x^*\|^2$$

$$+ \eta^2 \frac{\eta L_s - 1}{2}\|\nabla_{HT}^t\|^2 + \frac{\eta L}{2n}(1 - \frac{1}{\theta})\|x^t - \varphi_t^i\|^2$$

(7)

In the iterative process of the algorithm, we can use $\|\nabla_{HT}^t\|^2$ bound based on the $\left\langle g(x^t) - \nabla_F f(x^t), x^t - x^{t+1} \right\rangle$ due to Assumption 1 and $(1 - \frac{1}{\theta})\|x^t - \varphi_t^i\|^2$ due to definition 1. The complete proof steps can be found in the appendix.

**Remark 5** *In a memory-biased algorithm, the bias cannot be effectively eliminated. Therefore, (7) has worse bounds compared to (5), indicating poorer convergence.*

## 3.2 ZEROTH-ORDER HARD-THRESHOLDING

We should mention that for recursively biased oracles in zeroth-order method, lemma 2 is still holds. Therefore, we can use the same approach to study recursively biased zeroth-order hard-thresholding estimation. And

**Lemma 5** *If $g(x^t)$ is zeroth-order recursively biased, for any $\epsilon > 0$, there is*

$$\sum_{t=\nu s+1}^{\nu(s+1)-1} \|\mathbb{E}\langle \nabla_F f(x^{t-1}) - g(x^{t-1}), x^t - x^* \rangle\| \leq \sum_{t=\nu s+1}^{\nu(s+1)-1} \mathbb{E}[\frac{\nu\epsilon}{2}\|\nabla_F f(x_t) - g(x^t)\|^2 + \frac{\nu}{2\epsilon}\|x^{t+1} - x^t\|^2$$

$$+ \frac{\nu\epsilon}{2}\|\nabla_F f(x_t) - \hat{\nabla}_F f(x_t)\|^2 + \langle \nabla_F f(x^{t-1}) - \hat{\nabla}_F f(x^{t-1}), x^t - x^* \rangle + \left\langle \hat{\nabla}_F f\left(x^{vs}\right) - g(x^{vs}), x^{vs} - x^* \right\rangle].$$

(8)

**Remark 6** *Since $\nabla f(x^t) \neq \mathbb{E}_t g(x^t)$ due to the bias of zeroth-order estimation and $(\nabla_F f(x^{\nu s}) \neq g(x^t))_F$ due to the zeroth-order estimation cannot use the same $u$ in different $t$. This means that in recursively biased zeroth-order hard-thresholding algorithms, not only is the bias not partially canceled, but it also accumulates throughout the iterations. This implies that recursively biased is unlikely to achieve good convergence speed in the zeroth-order hard-thresholding setting.*

Now, we turn our attention to memory biased zeroth-order hard-threshodling. In the first-order discussion, we know that the bias produced by memory biased and hard-threshodling does not interact well. Therefore, we only need to study the MSE of $g(x)$. By doing so, we can understand the relationship between memory biased and zeroth-order biases.

**Lemma 6** *If $g(x^t)$ is memory biased estimation, for any $\theta > 1$ and $q < d$, we have*

$$
\begin{aligned}
\mathbb{E}\|g(x)\|_2^2 \leq{} & \frac{4q}{\theta^2 d}\mathbb{E}\|\nabla_{\mathcal{I}} f_{i^t}(x) - \nabla_{\mathcal{I}} f_{i^t}(x^*)\|^2 + (8 + \frac{4}{\theta^2})\frac{q}{d}\|\nabla_{\mathcal{I}} f_{i^t}(\varphi) - \nabla_{\mathcal{I}} f_{i^t}(x^*)\|^2 \\
& + 4\mathbb{E}\|\nabla_F F(x^*) + \tau_i\|^2,
\end{aligned}
\tag{9}
$$

*where $\tau_i = \mathbb{E}_{u, u^t, i}\left[\frac{1}{\theta}\left(s_i\left(x^t, u^t\right) - s_i\left(w^t, u^t\right)\right)\|u^t\|^2 u_F^t + s\left(w^t, u\right)\|u\|^2 u_F\right]$*

**Remark 7** *We point out that when $\tau_i$ is the bias introduced by zeroth-order estimation by the definition of $s_i(x, u)$. As $\theta$, which is the bias introduced by memory bias, increases, the bias introduced by zeroth-order correspondingly decreases. This indicates that the bias from memory can cancel out part of the bias from zeroth-order.*

### 3.3 CONCLUSION

From the above discussion, we know that the recursive-biased algorithm can partially cancel out the bias in first-order hard-thresholding, while the Memory-biased algorithm can cancel out part of the bias in zeroth-order hard-thresholding. This suggests that, compared to existing algorithms, we can design SARAH-HT and BVR-SZHT algorithms to achieve faster convergence rates.

## 4 BIASED HARD-THRESHOLDING ALGORITHM

In this chapter, we will provide a convergence analysis for the first-order algorithms SARAH-HT, BSVRG-HT, and BSAGA-HT, as well as the zeroth-order algorithm BVR-SZHT. Due to spatial limitations, the the algorithm for first-order will be placed in the appendix.

### 4.1 BIASED FIRST-ORDER HARD-THRESHOLDING ALGORITHM

**Theorem 1** *(Recursive-biased estimators) Let $g(x)$ be a recursive-biased gradient oracle parameterized by $\nu \geq 1$, which satisfies the $BMSE(M_1, M_2, \rho_M, \rho_F, m)$ property. Let $B_2 \overset{def}{=} \min\{\nu, 1/\rho_B\}$, $\Theta = \frac{M_1\rho_F + 2M_2}{\rho_M\rho_F}$ and $\rho = \min\{\rho_M, \rho_F\}$. Assume that each $f_i$ is $L_s-RSS$ and that $v_s-RSC$. For any stochastic hard-thresholding algorithms, we can establish the following:*

$$
\begin{aligned}
&\mathbb{E}[\alpha^{mK}(f(x^{mK}) - f(x^*)) + \frac{1}{2}\|x^{mK} - x^*\|^2] \\
&\leq \alpha^{-mK}\mathbb{E}[\delta'(f(x^0) - f(x^*)) + \frac{1}{2}\|x^0 - x^*\|^2] + \frac{\delta}{2\lambda'}\frac{\alpha^{mK} - \alpha^K}{\alpha^{mk}(\alpha - 1)}\|\nabla f(x^*)\|^2,
\end{aligned}
\tag{10}
$$

*where $\delta' = \frac{L_s^2}{v_s}\eta$, $\alpha = 1 + \delta/\kappa_s - 2\gamma_k$,*

$$
\delta = \frac{2\frac{1}{v_s} + 1 - (\frac{3B_2}{\epsilon} + 6B_2 L^2 \epsilon \Theta + \frac{1}{v_s}L_s\sqrt{2\Theta})\eta - 2\gamma_k}{L_s}
$$

**Remark 8** *The SARAH gradient estimator is recursively biased with parameters $\rho_B = 0$ and $\nu = m$, and it satisfies the BMSE property with parameters $M_1 = m, \rho_M = 1, \rho_F = 1$, and $M_2 = 0$.*

**Remark 9** *We note that if $f$ has a $k^*$-sparse unconstrained minimizer, which could happen in sparse reconstruction, or with overparameterized deep networks, then we would have $\|\nabla f(x^*)\| = 0$, and hence that part of the system error would vanish.*

**Theorem 2** *(Memory-biased estimation)Let $g(x)$ be a memory-biased gradient oracle, which satisfies the $BMSE(M_1, M_2, \rho_M, \rho_F, m)$ property. Let $\theta > 1$, and $B_1 \geq 0$, $\Theta = \frac{M_1\rho_F + 2M_2}{\rho_M\rho_F}$ and $\rho = \min\{\rho_M, \rho_F\}$. Assume that each $f_i$ is $L_s-RSS$ and that $v_s-RSC$. For any stochastic hard-thresholding algorithm, we can establish the following:*

$$
\begin{aligned}
&\mathbb{E}[\alpha^{mK}(f(x^{mK}) - f(x^*)) + \frac{1}{2}\|x^{mK} - x^*\|^2] \\
&\leq \alpha^{-mK}\mathbb{E}[\delta'(f(x^0) - f(x^*)) + \frac{1}{2}\|x^0 - x^*\|^2] + \frac{\delta}{2\lambda'}\frac{\alpha^{mK} - \alpha^K}{\alpha^{mk}(\alpha - 1)}\|\nabla f(x^*)\|^2,
\end{aligned}
\tag{11}
$$

*where $\delta' = \frac{L_s^2}{v_s}\eta, \alpha = 1 + \delta/\kappa_s - 2\gamma_k$*

$$
\delta = \frac{2\frac{L_s^2}{v_s} + 1 - (L_sB_1(1 - \frac{1}{\theta}) + (\frac{L_s^2}{v_s} + 1)L_s(2\sqrt{2\Theta} + 1))\eta - 2\gamma_k}{L_s}
$$

**Remark 10** *The B-SAGA gradient estimator is memory-biased with $B_1 = 2n(2n + 1)$, and it satisfies the $BMSE$ property with parameters $\rho_M = \frac{1}{2n}, m = 1, M_2 = 0, \rho_F = 1$, and*

$$
M_1 = \begin{cases} \frac{2n+1}{\theta^2} & \theta \in (0, 2] \\ (2n + 1)\left(1 - \frac{1}{\theta}\right)^2 & \theta > 2 \end{cases}.
$$

*The B-SVRG gradient estimator is memory-biased with $B_1 = 3m(m+1)$, and it satisfies the BMSE property with parameters $\rho_M = 1, M_2 = 0, \rho_F = 1$, and*

$$
M_1 = \begin{cases} \frac{3m(m+1)}{\theta^2} & \theta \in (0, 2] \\ 3m(m + 1)\left(1 - \frac{1}{\theta}\right)^2 & \theta > 2 \end{cases}.
$$

**Remark 11** *The convergence rate is $\alpha^{-1}$, which means that we can be using $\delta$ to compare it. In this way, we can find that SARAH-HT has a faster convergence rate than SVRG-HT.*

## 4.2 BIASED ZEROTH-ORDER HARD-THRESHOLDING ALGORITHM

**Theorem 3** *Assume the functions $\{f_i(\theta)\}_{i=1}^n$ satisfy the RSS condition Suppose that we run BVR-SZHT with random supports of size $s_2$, $q$ random directions, a learning rate of $\eta$, and $k$ coordinates kept at each iteration. We have: For BVR-SZHT algorithm, Let $\theta > 0$ Assume that each $f_i$ is $L_s-RSS$ and $v_s-RSC$ with $s = 2k + k^*$. we run BVR-SZHT with random supports of size $s_2$ random directions, a learning rate of $\eta$, and $k$ coordinates kept at each iteration. We have:*

$$
\begin{aligned}
\mathbb{E}\|x^m - x^*\|_2^2 \leq &(\beta^m - \frac{\beta^m - 1}{\beta - 1}\alpha(\eta v_s - \frac{4}{\theta^2} + \frac{\eta}{2}(1 - \frac{1}{\theta} - L_s\eta(1 - \frac{1}{\theta}))))\|x^0 - x^*\|_2^2 \\
&+ 4\frac{\beta^m - 1}{\beta - 1}\alpha(1 - \frac{1}{\theta})^2\mathbb{E}\|\nabla_F F(x^*)\|^2
\end{aligned}
\tag{12}
$$

*where $\beta = \alpha(1 - \frac{\eta v_s}{\theta}\sqrt{\frac{s}{d}} + \eta L_s 2(1 - \frac{1}{\theta})\sqrt{\frac{s}{d}} + \lambda\eta\sqrt{\frac{s}{d}} + \eta^2\frac{4s}{\theta^2 d}L_s 2 - 2\eta$,*
*$\alpha = \sqrt{1 + \left(K/k + \sqrt{(4 + K/k)K/k}\right)/2}$*

## 5 EXPERIMENTS

In this section, we conduct experiments on both the first-order and zeroth-order algorithms, focusing on adversarial attacks and sparse feature selection. The experiments are presented in two parts: first, we evaluate the effectiveness of different algorithms in sparse feature selection to highlight the

---

**Algorithm 1** Stochastic bias variance reduced Hard-Thresholding algorithm (BVR-SZHT)

---

**Input:** Learning rate $\eta$, maximum number of iterations $T$, initial point $x^0$, SVRG update frequency $m$, number of random directions $q$, and number of coordinates to keep at each iteration $k$, biased coefficient $\theta$.

**Output:** $x^T$.

    **for** $r = 1, \ldots, T$ **do**

        $x^{(0)} = x^{r-1}$;

        $\hat{\mu} = \frac{1}{n} \sum_{i=1}^{n} \hat{\nabla} f_i(x^{(0)})$;

        **for** $t = 0, 1, \ldots, m-1$ **do**

            Randomly sample $i_r \in \{1, 2, \ldots, n\}$;

            Compute ZO estimate $\hat{\nabla} f_{i_r}(x^{(r)}), \hat{\nabla} f_{i_r}(x^{(0)})$ with the same direction $u$;

            $\bar{x}^{(r+1)} = x^{(r)} - \eta(\frac{1}{\theta}(\hat{\nabla} f_{i_r}(x^{(r)}) - \hat{\nabla} f_{i_r}(x^{(0)})) + \hat{\mu}))$;

            $x^{(r+1)} = \mathcal{H}_k(\bar{x}^{(r+1)})$;

        **end for**

        $x^r = x^{(t')}$, random $t' \in [m-1]$

    **end for**

---

advantages of BVR-SZHT and SARAH-HT. Then, we analyze black-box adversarial attacks as a real-world application scenario for zeroth-order algorithms. The ridge regression and sensitivity analysis experiments, previously conducted to validate parameter effects, are now provided in the appendix for reference. These supplementary experiments include detailed sensitivity analysis of the parameter $k$ in the first-order algorithms and the parameter $\mu$ in the zeroth-order algorithms, aimed at observing the bias cancellation effects under increased bias from hard thresholding and zeroth-order estimation. The performance of the algorithms will be evaluated in terms of the following three aspects:

- IFO: the iterative first-order oracle, i.e. number of calls to $f_i$.
- IZO: the iterative zeroth-order oracle, i.e. number of calls to $f_i$.
- NHT: the number of hard-thresholding operations.

**Black-box Adversarial Attacks** Adversarial attacks trick machine learning models by adding carefully designed subtle perturbations to inputs, leading to mispredictions. Black-box adversarial attacks occur when attackers can't access a model's internals and must deduce its behavior from inputs and outputs. The Black-box attack method is closer to real-world attack scenarios. Therefore, we consider a few-pixel universal adversarial attack scenarios and assume there is a well-trained classifier that can only be accessed as a black box. In this scenario, zeroth-order algorithms excel over first-order ones in black-box settings as they don't need model gradients, estimating them through output queries instead. As is usual in black-box adversarial attacks, we maximize the following Carlini-Wagner loss (Carlini & Wagner, 2017; Chen et al., 2017), which promotes the model the model to make incorrect predictions:

$$f_i(\omega) = \max\{F_{y_i}(\text{clip}(x_i + \omega)) - \max_{j \neq y_i} F_j(\text{clip}(x_i + \omega)), 0\},$$

where $F$ denotes a pre-trained model, $x_i$ is the $i$-th image (rescaled to have values in $[-0.5, 0.5]$) with true class $y_i$, clip denotes the clipping operation into $[-0.5, 0.5]$, $\omega$ is the universal perturbation that we seek to optimize, and each $F_j$ outputs the log-probability of image $x_i$ being of class $n$ as predicted by the model ($j \in \{1, .., J\}$, $J$ is the number of classes, similarly to (Chen et al., 2017; Huang et al., 2019)). We use the pre-trained model on the CIFAR-10 as the model $F$. It can be obtained from the supplementary material of (de Vazelhes et al., 2022). Similarly to Liu et al. (2018), we evaluate the algorithms on a dataset of $n = 10$ images from the test-set of the CIFAR-10 dataset(Krizhevsky & Hinton, 2009). We set $k = 60$, $\mu = 0.001$, $q = 10$, $s_2 = d = 3,072$, the number of inner iterations of the variance reduced algorithms to $m = 10$ and the bias coefficient $\frac{1}{\theta} = 0.65$. We check at each iteration the number of IZO, and we stop training if it exceeds 600. Finally, we grid-search the learning rate $\eta$ in $\{0.001, 0.005, 0.01, 0.05\}$ and select the one that minimizes the loss value for each algorithm. The training curves are presented in Figure 5. We can observe that BVR-SZHT achieved the lowest loss value and showed significant performance improvement compared to VR-SZHT in this tasks.

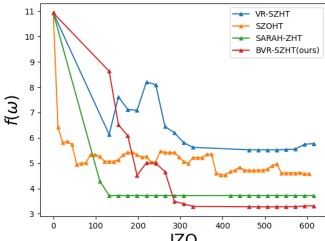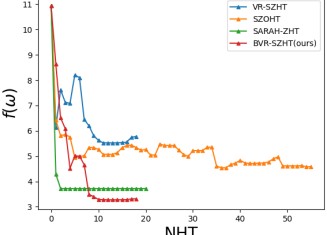

Figure 1: Loss values of ZO algorithms in black-box adversarial attack

**Sparse Feature Selection** Feature selection is a crucial step in reducing dimensionality and improving model interpretability, especially when dealing with high-dimensional biological datasets like scRNA-seq data. In our work, we applied several feature selection algorithms, BSVRG-HT, SAGA, and SARAH-HT, to efficiently select a subset of features that best represent the underlying biological signals. SAGA-LASSO, a popular approach for sparse logistic regression, uses the L1 penalty to encourage sparsity while leveraging stochastic optimization to solve large-scale problems efficiently. We conducted feature selection on scRNA-seq data and MINST/CIFAR-10 datasets from colorectal cancer cell lines. Following feature selection, we trained a deep neural network (DNN) to classify cell types based on the selected features. We optimized the hyperparameters, such as learning rates and batch sizes, for each feature selection algorithm to maximize the classification accuracy. The results of our experiments demonstrate the effectiveness of these methods in high-dimensional biological settings. BVRSZHT and SARAH both provided significant performance improvements in feature reduction while maintaining high accuracy. The selected features were subsequently used to train the DNN classifier, resulting in robust and interpretable predictions of cell type identities.

| Dataset | Algorithm | Accuracy | Num_Features | Selection_Time (s) |
|---------|-----------|----------|--------------|--------------------|
| Cancer | BVRSZHTn | 0.8850 | 2863 | 71.92 |
| Cancer | SAGA-LASSO | 0.9204 | 3470 | 645.07 |
| Cancer | BVRSZHT12 | 0.8673 | 2863 | 68.30 |
| Cancer | VRSZHT | 0.8496 | 2863 | 73.12 |
| Cancer | SARAH | 0.8938 | 2863 | 65.66 |
| CIFAR-10 | BVRSZHTn | 0.4575 | 1843 | 153.32 |
| CIFAR-10 | SAGA-LASSO | 0.5102 | 3053 | 5148.21 |
| CIFAR-10 | BVRSZHT12 | 0.5109 | 1843 | 152.18 |
| CIFAR-10 | VRSZHT | 0.5029 | 1843 | 150.75 |
| CIFAR-10 | SARAH | 0.5126 | 1843 | 153.08 |
| MNIST | BVRSZHTn | 0.9593 | 235 | 70.09 |
| MNIST | SAGA-LASSO | 0.9729 | 644 | 1131.67 |
| MNIST | BVRSZHT12 | 0.9563 | 235 | 70.43 |
| MNIST | VRSZHT | 0.9407 | 235 | 70.63 |
| MNIST | SARAH | 0.9616 | 235 | 64.00 |

Table 1: Reasult in sparse feature selesction

## 6 CONCLUSION

This paper investigates the interrelationship between gradient biases caused by different factors through the study of several specific algorithms. We found that the equivalent bias generated by hard-thresholding can be partially offset by the recursively biased in algorithms like SARAH, while the bias caused by zeroth-order gradients can be partially counteracted by the memory biased in BSVRG-type algorithms. Based on this theory, we designed the SARAH-HT algorithm and the BSVRG-HT algorithm, both of which demonstrate faster convergence compared to existing methods in first-order and zeroth-order hard-thresholding algorithms, respectively.

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

## A  NOTATIONS AND DEFINITIONS

Throughout this appendix, we will use the following notations:

- $\nabla f(x)$ denotes the gradient of $f$ at $x$.

- $g(x)$ denotes the gradient oracle of $f$ at $x$.

- $\hat{\nabla} f(x)$ denotes the zeroth-order of $f$ at $x$.

- $\boldsymbol{u}$ is the direction of zeroth-order.

- $\nabla_F f(\boldsymbol{x})$ denotes the gradient of $f$ at $\boldsymbol{x}$ in $F$.

- $[d]$ denotes the set of all integers between 1 and $d : \{1, \ldots, d\}$.

- $\boldsymbol{u}_i$ denotes the $i$-th coordinate of vector $\boldsymbol{u}$, and $\nabla_i f(\boldsymbol{x})$ the $i$-th coordinate of $\nabla f(\boldsymbol{x})$.

- $\| \cdot \|_0$ denotes the $\ell_0$ norm (which is not a proper norm).

- $\| \cdot \|$ denotes the $\ell_2$ norm.

- $\| \cdot \|_\infty$ denotes the maximum absolute component of a vector.

- $x \sim \mathcal{P}$ denotes that the random variable $\boldsymbol{X}$ (denoted as $\boldsymbol{x}$ ), of realization $\boldsymbol{x}$, follows a probability distribution $\mathcal{P}$ (we abuse notation by denoting similarly a random variable and its realization).

- $x_1, \ldots, x_n \overset{\text{i.i.d}}{\sim} \mathcal{P}$ denotes that we draw $n$ i.i.d. samples of a random variable $x$, each from the distribution $\mathcal{P}$.

- $P(x)$ denotes the value of the probability of $x$ according to its probability distribution.

- $\mathbb{E}_{x \sim \mathcal{P}}$ (or simply $\mathbb{E}_x$ if there is no possible confusion) to denote the expectation of $\boldsymbol{x}$ which follows the distribution $\mathcal{P}$.

- We denote by $\text{supp}(x)$ the support of a vector $\boldsymbol{x}$, that is the set of its non-zero coordinates.

- $|F|$ the cardinality (number of elements) of a set $F$.

- All the sets we consider are subsets of $[d]$. So for a given set $F$, $F^c$ denotes the complement of $F$ in $[d]$

- $\mathcal{S}^d(R)$ (or $\mathcal{S}^d(R)$ for simplicity if $R = 1$ ) denotes the $d$-sphere of radius $R$, that is $\mathcal{S}^d(R) = \{u \in \mathbb{R}^d / \|u\| = R\}$.

- $\mathcal{U}\left(\mathcal{S}^d\right)$ the uniform distribution on that unit sphere.

- $\beta(d)$ is the surface area of the unit $d$-sphere defined above.

- $\mathcal{S}^d_S$ denotes a set that we call the restricted $d$-sphere on $S$, described as: $\{\boldsymbol{u}_S / \boldsymbol{u} \in \{\boldsymbol{v} \in \mathbb{R}^d / \|v_S\| = 1\}\}$, that is the set of unit vectors supported by $S$.

- $\mathcal{U}\left(\mathcal{S}^d_S\right)$ denotes the uniform distribution on that restricted sphere above.

- We denote by $\boldsymbol{u}_F$ (resp. $\nabla_F f(\boldsymbol{x})$ ) the hard-thresholding of $\boldsymbol{u}$ (resp. $\nabla_F f(\boldsymbol{x})$ ) over the support $F$, that is, a vector which keeps $\boldsymbol{u}$ (resp. $\nabla f(\boldsymbol{x})$ ) untouched for the set of coordinates in $F$, but sets all other coordinates to 0 .

- $\binom{[d]}{s}$ denotes the set of all subsets of $[d]$ that contain $s$ elements: $\binom{[d]}{s} = \{S : |S| = s, S \subseteq [d]\}$.

- $\mathcal{U}\left(\binom{[d]}{s}\right)$ denotes the uniform distribution on the set above.

- $I$ denotes the identity matrix $I_{d \times d}$.

- $I_S$ denotes the identity matrix with 1 on the diagonal only at indices belonging to the support $S : I_{i,i} = 1$ if $i \in S$, and 0 elsewhere.

- $S \ni e$ denotes that set $S$ contains the element $e$.

- $(\boldsymbol{u}_i)_{i=1}^n$ denotes the $n$-uple of elements $\boldsymbol{u}_1, \ldots, \boldsymbol{u}_n$.

# B  LEMMA

For convenience proof, we need to divide the biased variance reduce algorithm into two parts. memorization biased part and iteration biased part.

## B.1  PROOF OF LEMMA 1:

By the definition of $\gamma_k$

$$\left\langle x^t - x^{t+1} - \eta g(x^t), x - x^{t+1} \right\rangle \leq \gamma_k \|x^{t+1} - x\|^2$$

$$\eta \left\langle g(x^t), x^{t+1} - x \right\rangle + \left\langle x^t - x^{t+1}, x - x^{t+1} \right\rangle \leq \gamma_k \|x^{t+1} - x\|^2$$

$$\eta \left\langle g(x^t), x^{t+1} - x \right\rangle + \frac{1}{2}\|x^t - x^{t+1}\|^2 + \frac{1}{2}\|x^{t+1} - x\|^2 - \frac{1}{2}\|x^t - x\|^2 \leq \gamma_k \|x^{t+1} - x\|^2$$

$$(\gamma_k - \frac{1}{2})\|x^{t+1} - x\|^2 + \frac{1}{2}\|x^t - x\|^2 - \frac{1}{2}\|x^t - x^{t+1}\|^2 \geq \eta \left\langle \hat{\nabla} f(x), x^{t+1} - x \right\rangle$$

$$\tag{13}$$

## B.2  PROOF OF LEMMA 2 , 3 AND 5:

**Proof of lemma 2:** From the RSS-condition:

$$\eta(f(x^t) - f(x^*)) \leq \eta \left\langle \nabla_F f(x^t), x^t - x^* \right\rangle$$

$$= \eta \left\langle \nabla_F f\left(x^t\right) - g(x^t), x^t - x^* \right\rangle + \eta \left\langle g(x^t), x^t - x^* \right\rangle.$$

$$\overset{(13)}{\leq} \eta \mathbb{E}[\left\langle \nabla_F f\left(x^t\right) - g(x^t), x^t - x^* \right\rangle + (\gamma_k - \frac{1}{2})\|x^{t+1} - x^*\|^2 + \frac{1}{2}\|x^t - x^*\|^2 - \frac{1}{2}\|x^t - x^{t+1}\|^2]$$

**Proof of lemma 3:** For $\mathbb{E}\left\langle \nabla_F f\left(x^t\right) - g(x^t), x^t - x^* \right\rangle$, we have:

$$\mathbb{E}\left\langle \nabla_F f\left(x^t\right) - g(x^t), x^t - x^* \right\rangle \overset{(1)}{=} \mathbb{E}\left\langle \nabla_F f\left(x^t\right) - \mathbb{E}_t g(x^t), x^t - x^* \right\rangle \overset{(2)}{\leq} \mathbb{E}\left\langle \nabla_F f\left(x^{t-1}\right) - g(x^{t-1}), x^t - x^* \right\rangle$$

We can pass the conditional expectation $\mathbb{E}_{t-1}$ into the second inner-product in (1) because $x_{t-1}$ is independent of $j_{t-1}$. Inequality (2) uses the definition of a recursively biased gradient oracle. This is a recursive inequality, and expanding the recursion gives

$$\mathbb{E}\left\langle \nabla_F f\left(x^{t-1}\right) - g(x^{t-1}), x^t - x^* \right\rangle$$

$$= \mathbb{E}\left\langle \nabla_F f\left(x^{t-1}\right) - g(x^{t-1}), x^t - x^{t-1} \right\rangle + \mathbb{E}\left\langle \nabla_F f\left(x^{t-1}\right) - g(x^{t-1}), x^{t-1} - x^* \right\rangle$$

$$\leq \sum_{l=vs+1}^{t-1} \mathbb{E}\left\langle \nabla_F f\left(x^{l-1}\right) - g(x^{l-1}), x^l - x^{l-1} \right\rangle + \mathbb{E}\left\langle \nabla_F f\left(x^{vs}\right) - g(x^{vs}), x^{vs} - x^* \right\rangle$$

$$\overset{1}{=} \sum_{l=vs+1}^{t-1} \mathbb{E}\left\langle \nabla_F f\left(x^{l-1}\right) - g(x^{l-1}), x^l - x^{l-1} \right\rangle$$

Equation 1 is due to the fact that $\nabla_F f\left(x^{l-1}\right) = g(x^{l-1})$. Taking the absolute value and summing this from $t = \nu s + 1$ to $t = \nu(s + 1) - 1$

$$\sum_{t=\nu s+1}^{\nu(s+1)-1} \left\| \mathbb{E}\left\langle \nabla f\left(x^{t-1}\right) - g(x^{t-1}), x^t - x^* \right\rangle \right\|$$

$$\leq \sum_{t=\nu s+1}^{\nu(s+1)-1} \sum_{\ell=\nu s+1}^{t-1} \mathbb{E}\left[ \frac{\epsilon}{2} \left\| \nabla f\left(x^\ell\right) - g(x^\ell) \right\|^2 + \frac{1}{2\epsilon} \left\| x^{\ell+1} - x^\ell \right\|^2 \right] \tag{14}$$

$$\leq \nu \sum_{t=\nu s+1}^{\nu(s+1)-1} \mathbb{E}\left[ \frac{\epsilon}{2} \left\| \nabla f\left(x_t\right) - g(x^t) \right\|^2 + \frac{1}{2\epsilon} \left\| x^{t+1} - x^t \right\|^2 \right]$$

Summing this inequality from $s = 0$ to $s = S$ completes the proof.

Lemma 5 can be easily proof by lemm 3

### B.3 PROOF OF LEMMA 4:

**Proof:** By assumption, $1 - \frac{1}{\theta} \geq 0$, so we can apply RSC to obtain:

$$\frac{\eta}{\theta}(f(x^t) - f(x^*)) + \frac{\eta}{n}(1 - \frac{1}{\theta})(\sum_{i=1}^{n} f_i(\varphi_t^i) - f_i(x^*))$$

$$\leq \frac{n}{\theta}\left\langle \nabla f(x^t), x^t - x^* \right\rangle + \frac{\eta}{n}(1 - \frac{1}{\theta})\sum_{i=1}^{n}\left\langle \nabla f_i(\varphi_t^i), \varphi_t^i - x^* \right\rangle$$

$$= \frac{n}{\theta}\left\langle \nabla f(x^t), x^t - x^* \right\rangle + \frac{\eta}{n}(1 - \frac{1}{\theta})\sum_{i=1}^{n}\left\langle \nabla f_i(\varphi_t^i), \varphi_t^i - x^t \right\rangle + \frac{\eta}{n}(1 - \frac{1}{\theta})\sum_{i=1}^{n}\left\langle \nabla f_i(\varphi_t^i), x^t - x^* \right\rangle \tag{15}$$

Since $g(x)$ is memory-biased,

$$\frac{1}{\theta}\nabla f(x^t) + \frac{1}{n}(1 - \frac{1}{\theta})\sum_{i=1}^{n} \nabla f_i(\varphi_t^i) = \mathbb{E}_t[g(x^t)]$$

Therefore,

$$\frac{n}{\theta}\left\langle \nabla f(x^t), x^t - x^* \right\rangle + \frac{\eta}{n}(1 - \frac{1}{\theta})\sum_{i=1}^{n}\left\langle \nabla f_i(\varphi_t^i), x^t - x^* \right\rangle$$

$$= \mathbb{E}[\eta\left\langle g(x^t), x^t - x^* \right\rangle] \tag{16}$$

$$= \mathbb{E}[\eta\left\langle g(x^t), x^t - x^{t+1} \right\rangle] + \mathbb{E}[\eta\left\langle g(x^t), x^{t+1} - x^* \right\rangle]$$

$$\leq \mathbb{E}[\eta\left\langle g(x^t), x^t - x^{t+1} \right\rangle + (\gamma_k - \frac{1}{2})\|x^{t+1} - x^*\|^2 + \frac{1}{2}\|x^t - x^*\|^2 - \frac{1}{2}\|x^t - x^{t+1}\|^2]$$

The inequality is due to lemma 1 with $x = x^*$. Combining these two inequalities, we have shown:

$$\frac{n}{\theta}(f(x^t) - f(x^*)) + \frac{\eta}{n}(1 - \frac{1}{\theta})(\sum_{i=1}^{n} f_i(\varphi_t^i) - f_i(x^*))$$

$$\leq \mathbb{E}[\eta\left\langle g(x^t), x^t - x^{t+1} \right\rangle - (\frac{1}{2} - \gamma_k)\|x^{t+1} - x^*\|^2 + \frac{1}{2}\|x^t - x^*\|^2$$

$$- \frac{1}{2}\|x^t - x^{t+1}\|^2 + \frac{\eta}{n}(1 - \frac{1}{\theta})\sum_{i=1}^{n}\left\langle \nabla f_i(\varphi_t^i), x^t - x^* \right\rangle]$$

$$\leq \mathbb{E}[\eta\left\langle g(x^t) - \nabla f(x^t), x^t - x^{t+1} \right\rangle + \left\langle \nabla f(x^t), x^t - x^{t+1} \right\rangle - (\frac{1}{2} - \gamma_k)\|x^{t+1} - x^*\|^2 + \frac{1}{2}\|x^t - x^*\|^2$$

$$- \frac{1}{2}\|x^t - x^{t+1}\|^2 + \frac{\eta}{n}(1 - \frac{1}{\theta})\sum_{i=1}^{n}\left\langle \nabla f_i(\varphi_t^i), x^t - x^* \right\rangle]$$

$$\leq \mathbb{E}[\eta\left\langle g(x^t) - \nabla f(x^t), x^t - x^{t+1} \right\rangle + f(x^t) - f(x^{t+1}) - (\frac{1}{2} - \gamma_k)\|x^{t+1} - x^*\|^2 + \frac{1}{2}\|x^t - x^*\|^2$$

$$+ (\frac{\eta L_s - 1}{2})\|x^t - x^{t+1}\|^2 + \frac{\eta}{n}(1 - \frac{1}{\theta})\sum_{i=1}^{n}\left\langle \nabla f_i(\varphi_t^i), x^t - x^* \right\rangle] \tag{17}$$

Here, organize the equation. Using RSS condition, we have:

$$0 \leq -\eta\mathbb{E}[f(x^{t+1}) - f(x^*)] + \eta \left\langle g(x^t) - \nabla f(x^t), x^t - x^{t+1}\right\rangle - (\frac{1}{2} - \gamma_k)\|x^{t+1} - x^*\|^2 + \frac{1}{2}\|x^t - x^*\|^2$$

$$+ (\frac{\eta L_s - 1}{2})\|x^t - x^{t+1}\|^2 + \eta(1 - \frac{1}{\theta})\left(f(x^t) - \frac{1}{n}\sum_{i=1}^{n} f_i(\varphi_t^i) + \frac{1}{n}\sum_{i=1}^{n}\left\langle\nabla f_i(\varphi_t^i), \varphi_t^i - x^t\right\rangle\right)$$

$$\leq -\eta\mathbb{E}[f(x^{t+1}) - f(x^*)] + \eta \left\langle g(x^t) - \nabla f(x^t), x^t - x^{t+1}\right\rangle - (\frac{1}{2} - \gamma_k)\|x^{t+1} - x^*\|^2 + \frac{1}{2}\|x^t - x^*\|^2$$

$$+ (\frac{\eta L_s - 1}{2})\|x^t - x^{t+1}\|^2 + \frac{\eta L}{2n}(1 - \frac{1}{\theta})\|x^t - \varphi_t^i\|^2$$

$$\tag{18}$$

## B.4 PROOF OF LEMMA 6

By the definition of $g$, we can verify the second claim as:

$$\mathbb{E}\|g(x)\|_2^2 = \mathbb{E}\|\frac{1}{\theta}\hat{\nabla}_{\mathcal{I}}f_i\left(x^t, u_k\right) - \frac{1}{\theta}\hat{\nabla}_{\mathcal{I}}f_i\left(w_k, u_k\right) + \hat{\nabla}_{\mathcal{I}}f\left(w_k, u\right)\|$$

$$\leq \frac{4}{\theta^2}\mathbb{E}\|u_{F,i}u^T\nabla_{\mathcal{I}}f_{i^t}(x) - u_{F,i}u_i^T\nabla_{\mathcal{I}}f_{i^t}(x^*)\|^2 + 4\mathbb{E}\|\nabla_F F(x^*) + \tau\|^2$$

$$+ 4\|u_F u^T(\nabla_{\mathcal{I}}f_i(\varphi) - \nabla_{\mathcal{I}}f_i(x^*)) - \frac{1}{\theta}u_{F,i}u_i^T(\nabla_{\mathcal{I}}f_i(\varphi) - \nabla_{\mathcal{I}}f_i(x^*))\|^2$$

$$+ \mathbb{E}\|u_F\|^2\|u\|^2\|(\nabla_{\mathcal{I}}f_i(\varphi) - \nabla_{\mathcal{I}}f_i(x^*)) - \mathbb{E}(\nabla_{\mathcal{I}}f_i(\varphi) - \nabla_{\mathcal{I}}f_i(x^*))\|^2$$

$$\leq \frac{4s}{\theta^2 d}\mathbb{E}\|\nabla_{\mathcal{I}}f_{i^t}(x) - \nabla_{\mathcal{I}}f_{i^t}(x^*)\|^2 + (8 + \frac{4}{\theta^2})\frac{q}{d}\|\nabla_{\mathcal{I}}f_{i^t}(\varphi) - \nabla_{\mathcal{I}}f_{i^t}(x^*)\|^2$$

$$+ 4\mathbb{E}\|\nabla_F F(x^*) + \tau\|^2$$

$$\tag{19}$$

## B.5 OTHER LEMMAS

**Lemma 7** (MSE bound) *Suppose that the stochastic gradient oracle $\hat{\nabla}_F$ satisfies the BMSE$(M_1, M_2, \rho_M, \rho_F, m)$ property, let $\rho = min\{\rho_M, \rho_F\}$, and let $\sigma_s$ be any sequence satisfying $\sigma_s(1-\rho)^{m(s-l)} \leq \sigma_l(1-\frac{\rho}{2})^{m(s-l)}$. For convenience, define $\Theta = \frac{M_1\rho_F + 2M_2}{\rho_M\rho_F}$ and $\mathcal{I} = \mathcal{I}_k + \mathcal{I}_{k+1}$. The MSE of the gradient oracle is bounded as*

$$\sum_{t=ms}^{m(s+1)-1}\mathbb{E}[\|\nabla_F f(x^t) - gf(x^t)\|^2] \leq \Theta L_s^2 \sum_{t=ms}^{m(s+1)-1}\mathbb{E}[\|x^{t+1} - x^t\|^2]$$

**Proof** First, we derive a bound on the sequence $f_{ms}$ arising in the BMSE property. Summing this sequence from $s = 0$ to $s = S$.

$$\sigma_s f_{ms} \leq \sum_{l=0}^{s}\frac{M_2\sigma_s(1-\rho_F)^{m(s-l)}}{n}\sum_{k=ms}^{m(s+1)-1}\sum_{i=1}^{n}\mathbb{E}[\|\nabla_F f_i(x^{k+1}) - \nabla_F f_i(x^t)\|^2]$$

$$\leq \sum_{l=0}^{s}\frac{M_2\sigma_l(1-\frac{\rho_F}{2})^{m(s-l)}}{n}\sum_{k=ms}^{m(s+1)-1}\sum_{i=1}^{n}\mathbb{E}[\|\nabla_F f_i(x^{k+1}) - \nabla_F f_i(x^t)\|^2]$$

$$\leq \sum_{l=0}^{\infty}(1-\frac{\rho_F}{2})^l\frac{M_2\sigma_s}{n}\sum_{k=ms}^{m(s+1)-1}\sum_{i=1}^{n}\mathbb{E}[\|\nabla_F f_i(x^{k+1}) - \nabla_F f_i(x^t)\|^2]$$

$$= \frac{2M_2\sigma_s}{n\rho_F}\sum_{k=ms}^{m(s+1)-1}\sum_{i=1}^{n}\mathbb{E}[\|\nabla_F f_i(x^{k+1}) - \nabla_F f_i(x^t)\|^2]$$

$$\tag{20}$$

For $\mathcal{M}_{ms}$ similarly:

$$\sigma_s \mathcal{M}_{ms} \leq \sigma_s \left( f_{ms} + \frac{M_1}{n} \sum_{k=ms}^{m(s+1)-1} \sum_{i=1}^{n} \mathbb{E}[\|\nabla_F f_i(x^{k+1}) - \nabla_F f_i(x^k)\|^2] \right) + (1 - \rho_M)^m \sum_{s=1}^{S} \sigma_s \mathcal{M}_{m(s-1)}$$

$$\leq \sigma_s \left( \frac{M_1 \rho_F + 2M_2}{n\rho_F} \sum_{k=ms}^{m(s+1)-1} \sum_{i=1}^{n} \mathbb{E}[\|\nabla_F f_i(x^{k+1}) - \nabla_F f_i(x^k)\|^2] \right) + (1 - \frac{\rho_M}{2})^m \sum_{s=1}^{S} \rho_{s-1} \mathcal{M}_{m(s-1)}$$

$$\leq \left( \sum_{l=0}^{\infty} (1 - \frac{\rho_M}{2})^{ml} \right) \sigma_s \left( \frac{M_1 \rho_F + 2M_2}{n\rho_F} \sum_{k=ms}^{m(s+1)-1} \sum_{i=1}^{n} \mathbb{E}[\|\nabla_F f_i(x^{k+1}) - \nabla_F f_i(x^k)\|^2] \right)$$

$$\leq \frac{2\sigma_s \Theta}{n} \sum_{k=ms}^{m(s+1)-1} \sum_{i=1}^{n} \mathbb{E}[\|\nabla_F f_i(x^{k+1}) - \nabla_F f_i(x^k)\|^2]$$

$$\leq 2\Theta L_s^2 \sigma_s \sum_{k=ms}^{m(s+1)-1} \sum_{i=1}^{n} \mathbb{E}[\|x^{k+1} - x^k\|^2]$$

$$\tag{21}$$

From assumption 1, we have the conclusion.

**Lemma 8** $\theta \geq 1$ and that $f$ is $L_S - RSC$ with $\mu \geq 0$. For any $\nabla > 0$, the following inequality holds:

$$f(x^{t+1}) - f(x^*) \leq \frac{1}{2\lambda'} \|\nabla f(x^*)\|^2 + \frac{\lambda'}{2} \|x^{t+1} - x^*\|^2 + \frac{L_s}{2} \|x^{t+1} - x^t\|^2 + (\frac{L_s/\lambda' - v_s}{2}) \|x^t - x^*\|^2 \tag{22}$$

**Proof:** from RSS and RSC condition, when $\eta < \frac{1}{L_s}$, we have:

$$f(x^*) \geq f(x^t) + \langle \nabla f(x^t), x^* - x^t \rangle + \frac{v_s}{2} \|x^t - x^*\|^2 \tag{23}$$

and

$$f(x^{t+1}) \leq f(x^t) + \langle \nabla f(x^t), x^{t+1} - x^t \rangle + \frac{L_s}{2} \|x^{t+1} - x^t\|^2 \tag{24}$$

From (23) and (24), we have:

$$f(x^{t+1}) - f(x^*) \leq \langle \nabla f(x^t), x^{t+1} - x^* \rangle + \frac{L_s}{2} \|x^{t+1} - x^t\|^2 - \frac{v_s}{2} \|x^t - x^*\|^2$$

$$\leq \frac{\lambda'}{2} \|\nabla f(x^t) - \nabla f(x^*)\|^2 + \frac{\lambda'}{2} \|\nabla f(x^*)\|^2 + \frac{1}{2\lambda'} \|x^{t+1} - x^*\|^2 + \frac{L_s}{2} \|x^{t+1} - x^t\|^2 - \frac{v_s}{2} \|x^t - x^*\|^2$$

$$\leq \frac{\lambda'}{2} \|\nabla f(x^*)\|^2 + \frac{1}{2\lambda'} \|x^{t+1} - x^*\|^2 + \frac{L_s}{2} \|x^{t+1} - x^t\|^2 + (\frac{\lambda' L_s - v_s}{2}) \|x^t - x^*\|^2$$

$$\tag{25}$$

**Lemma 9** *From RSS-condition, we have:*

$$f(x^{t+1}) - f(x^t) \leq \langle \nabla f(x^t), x^{t+1} - x^t \rangle + \frac{L_s}{2} \|x^{t+1} - x^t\|^2$$

$$\leq \langle \nabla f(x^t) - g(x^t), x^{t+1} - x^t \rangle + \langle g(x^t), x^{t+1} - x^t \rangle + \frac{L_s}{2} \|x^{t+1} - x^t\|^2 \tag{26}$$

*From lemma 1 , let $x = x^t$, we have:*

$$f(x^{t+1}) - f(x^t) \leq \langle \nabla f(x^t) - g(x^t), x^{t+1} - x^t \rangle - \frac{2 - 2\gamma_k - \eta L_s}{2\eta} \|x^t - x^{t+1}\|^2$$

**Lemma 10** *(Proofed in (Liu & Foygel Barber, 2020)) The relative concavity of hard-thresholding is given by*

$$\gamma_k = \frac{\sqrt{\frac{k^*}{k}}}{2}$$

**Lemma 11** *Let $b \in \mathbb{R}^d$ be an arbitrary vector and $\boldsymbol{b} \in \mathbb{R}^d$ be an arbitrary vector and $x \in \mathbb{R}^d$ be any $K$-sparse signal. For any $k \geq K$, we have the following bound:*

$$\|\mathcal{H}_k(\boldsymbol{b}) - x\|_2 \leq \sqrt{\nu}\|\boldsymbol{b} - x\|_2, \nu = \sqrt{1 + \left(K/k + \sqrt{(4 + K/k)K/k}\right)}/2$$

**Lemma 12** *(Proofed by de Vazelhes et al. (2022)) Let $F$ be a sub section of $[d]$, of size $s$, with $(s, d) \in \mathbb{N}_*^2$. We have the following:*

$$\mathbb{E}_{\boldsymbol{u} \sim \mathcal{U}(\mathcal{S}^d)} \|\boldsymbol{u}_F\| \leq \sqrt{\frac{s}{d}}$$

$$\mathbb{E}_{\boldsymbol{u} \sim \mathcal{U}(\mathcal{S}^d)} \|\boldsymbol{u}_F\|^2 = \frac{s}{d} \tag{27}$$

$$\mathbb{E}_{\boldsymbol{u} \sim \mathcal{U}(\mathcal{S}^d)} \|\boldsymbol{u}_F\|^4 = \frac{(s+2)s}{(d+2)d}$$

**Lemma 13** *Let the random vector $u$ drawn from the multivariate Gaussian distribution $\mathcal{N}(0, I_d)$. For the $L$-smooth function $f_i$ and any $x \in \mathbb{R}^d, i \in [n]$, the estimator in Eq.(2) satisfies:*

$$\hat{\nabla} f_i(x, u) = u_F u^\top \nabla_F f_i(x) + \frac{Lv}{2} u s_i(x, u)\|u\|^2, \tag{28}$$

*and its expectation w.r.t. $u$ is*

$$\mathbb{E}_u[g_{\mathcal{I},i}(x, u)] = \frac{1}{\theta}\nabla_F f(x^t) + (1 - \frac{1}{\theta})\nabla_F f(x^0) + \frac{Lv}{2}\tau_i(x, u),$$

**Proof of Lemma 13** For the RSS condition, we have the following Taylor expansion,

$$f_i(x + \mu u) = f_i(x) + \mu \langle \nabla f_i(x), u \rangle + \frac{\mu^2}{2} u^\top \nabla^2 f_i(x') u,$$

where $x' \in (x, x + vu)$. From the definition of $\nabla_F f_i(x)$, we have

$$g_{\mathcal{I},i}(x, u) = u_F \langle u, \nabla f_i(x) \rangle + \frac{v}{2} u^\top \nabla^2 f_i(x') u u_F$$

$$= u_F u^\top \nabla f_i(x) + \frac{Lv}{2} s_i(x, u)\|u\|^2 u_F$$

where the last equality employs the fact that $0 \preceq \nabla^2 f_i(x') \preceq L$ for any accessible $x'$, and the function $s_i(x, u)$ is confined to the range $[0, 1]$. Taking the expectation w.r.t. $u$ for $\hat{\nabla} f_i(x)$, we have

$$\mathbb{E}\left[\hat{\nabla}_{\mathcal{I}} f_i(x, u)\right] = \frac{f_i(x + \mu u) - f_i(x)}{\mu} u_F$$

$$= \mathbb{E}[u_F < u^\top, \nabla f_i(x) >] + \frac{Lv}{2}\mathbb{E}\left[s_i(x, u)\|u\|^2 u_F\right]$$

$$= \sqrt{\frac{s}{d}}\nabla f_i(x) + \frac{Lv}{2}\mathbb{E}\left[s_i(x, u)\|u\|^2 u_F\right].$$

Since $\left\|\mathbb{E}\left[s_i(x, u)\|u\|^2 u_F\right]\right\| \leq \mathbb{E}\left[\left\|s_i(x, u)\|u\|^2 u_F\right\|\right] \leq \mathbb{E}\left[\|\|u\|^2 u_F\|\right] = \mathbb{E}\left[\|u_F\|\right]$, with Eq.(27) $\mathbb{E}\left[\|u_F\|\right] \leq \sqrt{\frac{s}{d}}$, we then have $\left\|\mathbb{E}\left[s_i(x, u)\|u\|^2 u\right]\right\| \leq \sqrt{\frac{s}{d}}$. For the expected norm,

we have

$$\mathbb{E}\left[\left\|\hat{\nabla}f_i(x,u) - u_F u^T \nabla_F f_i(x^*)\right\|^2\right]$$

$$\overset{(4)}{=} \mathbb{E}\left[\left\|u_F u^\top \nabla_F f_i(x) + \frac{L\mu}{2}s_i(x,u)\|u\|^2 u_F - u_F u^T \nabla_F f_i(x^*)\right\|^2\right]$$

$$\leq \frac{L^2\mu^2}{2}\mathbb{E}\left[\|u\|^4\|u_F\|^2\right] + 2\mathbb{E}_u\|u\|^2\|u_F\|^2\mathbb{E}_i\|\nabla_F f_i(x) - \nabla_F f_i(x^*)\|^2$$

$$\leq \frac{L^2\mu^2 s}{2d} + 2\frac{q}{d}\mathbb{E}\|\nabla_F f_i(x) - \nabla_F f_i(x^*)\|^2$$

**Lemma 14** *Let $v$ be any vector in $\mathbb{R}^d$. For the random vector $u$ with the Gaussian distribution, i.e., $u \sim \mathcal{N}(0, I_d)$, we have*

$$\mathbb{E}_u\left[\left\|u_F u^\top v\right\|^2\right] = \frac{q}{d}\|v\|^2$$

**Proof.**

$$\mathbb{E}_u\left[\left\|u_F u^\top v\right\|^2\right] = \mathbb{E}_u\left[\|u_F\|^2\|u\|^2\|v\|^2\right] \quad = \frac{q}{d}\|v\|^2$$

**Lemma 15** *For the $L$-smooth function $f_i, i \in [n]$, the expected value of $g^t$ defined in Eq.(13) is*

$$\mathbb{E}_{u,u^t,i}\left[g^t\right] = \frac{1}{\theta}\sqrt{\frac{s}{d}}\nabla_F f\left(x^t\right) + (1 - \frac{1}{\theta})\sqrt{\frac{s}{d}}\nabla_F f\left(w^t\right) + \frac{Lv}{2}\tau_{i,k},$$

*where $\|\tau_{i,k}\| = \left\|\mathbb{E}_{u,u^t,i}\left[\frac{1}{\theta}\left(s_i\left(x^t,u^t\right) - s_i\left(w^t,u^t\right)\right)\|u^t\|^2 u_F^t + s\left(w^t,u\right)\|u\|^2 u_F\right]\right\|$ with the norm $\|\tau_{i,k}\| \leq 2\sqrt{q/d}$.*

## C  PROOF FOR FIRST-ORDER ALGORITHM

**Proof of Theorem 1:**
from Lemma 8,9 and 2, we have:

$$\mathbb{E}[(\eta + \delta)f(x^{t+1}) - f(x^*)] + \delta'\mathbb{E}[(f(x^{t+1}) - f(x^t))]$$

$$\leq \delta'\left\langle g(x^t) - \nabla_F f(x^t), x^t - x^{t+1}\right\rangle - (\frac{1 + \delta\lambda' - 2\gamma_k}{2})\|x^{t+1} - x^*\|^2 + \frac{1 + \delta L_s/\lambda' - \delta v_s}{2}\|x^t - x^*\|^2$$

$$+ \frac{\delta}{2\lambda'}\|\nabla_F f(x^*)\|^2 - (\frac{1 - \delta L_s}{2} + \delta'\frac{2 - 2\gamma_k - \eta L_s}{2\eta})\|x^t - x^{t+1}\|^2 + \eta\left\langle\nabla_F f\left(x^{t-1}\right) - g(x^{t-1}), x^t - x^*\right\rangle.$$

$$(29)$$

Let $\lambda' = \frac{1}{\kappa_s}$, $\alpha = 1 + \delta/\kappa_s - 2\gamma_k$. Multiplying (29) by $\alpha^t$, and summing over the epoch $t = ms$ to $t = m(s+1) - 1$, we have:

$$\sum_{t=ms}^{m(s+1)-1} \alpha^t\mathbb{E}[(\eta + \delta)f(x^{t+1}) - f(x^*) + \delta'(f(x^{t+1}) - f(x^t))]$$

$$\leq \frac{1}{2}\|x^{ms} - x^*\|^2 - \frac{1}{2}\alpha^{m(s+1)}\|x^{m(s+1)} - x^*\|^2 + \frac{\eta\kappa_s}{2}\sum_{t=ms}^{m(s+1)-1}\alpha^t\|\nabla_F f(x^*)\|^2$$

$$(30)$$

$$+ \sum_{t=ms}^{m(s+1)-1}\alpha^t\mathbb{E}\left[\delta\left\langle\nabla_F f\left(x^{t-1}\right) - g(x^{t-1}), x^t - x^*\right\rangle + \delta'\left\langle g(x^t) - \nabla_F f(x^t), x^t - x^{t+1}\right\rangle\right.$$

$$\left. - (\frac{1 - \delta L_s}{2} + \delta'\frac{2 - 2\gamma_k - \delta L_s}{2\eta})\|x^t - x^{t+1}\|^2\right]$$

Here, we have:

$$\alpha^t < \alpha^T \le \alpha^{T-1} \lim_{m \to \infty} (1 + \frac{1}{m})^m = e\alpha^T$$

where $e$ is Euler's number. We use Lemma 7 with $\sigma_k = \alpha^T$ to bound the MSE. Recall $\rho = \min\{\rho_M, \rho_F\}$ and $\eta/\kappa_s - 2\gamma_k \le \frac{\rho}{2}$. This choice for $\sigma_s$ satisfies the conditions of Lemma 7 because $\alpha^{mk}(1-\rho)^{mk} \le \alpha^{m(k-1)}(1-\rho/2)^{mk}$. We use the fact that the gradient oracle is recursively biased to bound the trem $\left\langle \nabla_F f\left(x^{t-1}\right) - \widetilde{\nabla}_F f(x^{t-1}), x^t - x^* \right\rangle$ and $\left\langle g(x^t) - \nabla_F f(x^t), x^t - x^{t+1} \right\rangle \le \|g(x^t) - \nabla_F f(x^t)\| \cdot \|x^t - x^{t+1}\|$. After that, summing the inequality from $s = 0$ to $s = S - 1$, $T = $ Then:

$$\sum_{s=0}^{S-1} \alpha^{ms} \sum_{t=ms}^{m(s+1)-1} \mathbb{E}[(\eta + \delta)f(x^{t+1}) - f(x^*) + \delta'(f(x^{t+1}) - f(x^t))]$$

$$\le \frac{1}{2}\|x^t - x^*\|^2 - \frac{1}{2}\alpha^{mS}\|x^{t+1} - x^*\|^2 + \frac{\delta\kappa_s}{2}\sum_{t=0}^{mS}\alpha^t\|\nabla_F f(x^*)\|^2 \tag{31}$$

$$+ C\sum_{s=0}^{S-1} \alpha^{ms} \sum_{t=ms}^{m(s+1)-1} \mathbb{E}\|x^t - x^{t+1}\|^2$$

Where $C = e\left(\frac{3B_2\eta}{2\epsilon} + 3B_2\eta L^2\epsilon\Theta + \delta'\sqrt{2\Theta}L_s - \left(\frac{1-\delta L_s}{2} + \delta'\frac{2-2\gamma_k-\eta L_s}{2\eta}\right)\right). \delta' = \frac{1}{v_s}\eta$. We see $C$ is zero if:

$$\delta = \frac{2\frac{1}{v_s} + 1 - \left(\frac{3B_2}{\epsilon} + 6B_2L^2\epsilon\Theta + \frac{1}{v_s}L_s\sqrt{2\Theta}\right)\eta - 2\gamma_k}{L_s}$$

Recalling that we have $\frac{1+2n\gamma_k}{n\kappa_s} \le \delta \le 2\frac{\gamma_k}{\kappa_s}$. So we have

$$\frac{1 + 2n\gamma_k}{n\kappa_s} \le \frac{2\frac{1}{v_s} + 1 - \left(\frac{3B_2}{\epsilon} + 6B_2L^2\epsilon\Theta + \frac{1}{v_s}L_s\sqrt{2\Theta}\right)\eta - 2\gamma_k}{L_s} \le 2\frac{\gamma_k}{\kappa_s}$$

That is:

$$\frac{\frac{2\gamma_k L_s}{\kappa_s} + 2\gamma_k - (2\frac{1}{v_s} + 1)}{\frac{3B_2}{\epsilon} + 6B_2L^2\epsilon\Theta + \kappa_s\sqrt{2\Theta}} \le \eta \le \frac{\frac{L_s\kappa_s + 2n\gamma_k L_s}{n\kappa_s} + 2\gamma_k - (2\frac{1}{v_s} + 1)}{\frac{3B_2}{\epsilon} + 6B_2L^2\epsilon\Theta + \kappa_s\sqrt{2\Theta}}$$

leaves So the step size in the theorem statement ensures $C = 0$ we have

$$(\eta + \delta + \delta')\sum_{t=0}^{T-1} \alpha^t\mathbb{E}[f(x^{t+1}) - f(x^*)] - \delta'\sum_{t=0}^{T-1} \alpha^t\mathbb{E}[f(x^t) - f(x^*)]$$

$$\le \frac{1 + \delta L_s/\lambda' - \delta v_s}{2}\|x^t - x^*\|^2 - \frac{1 + \delta L_s/\lambda' - \delta v_s}{2}\alpha^{mK}\|x^{t+1} - x^*\|^2 + \frac{\delta}{2\lambda'}\sum_{k=0}^{m(K-1)}\alpha^k\|\nabla_F f(x^*)\|^2 \tag{32}$$

Since $\delta' = \frac{L_s}{v_s}$ and $\eta < \frac{1}{L_s} < 1 + \frac{1}{m}$, we would like to show that $(1 + \delta + \delta') \ge \alpha\delta'$ so that the terms on the first line telescope.

$$\delta'\alpha^{mK}\mathbb{E}[f(x^{mK}) - f(x^*)] + \frac{1}{2}\alpha^{mK}\|x^{mK} - x^*\|^2$$

$$\le \delta'\mathbb{E}[f(x^0) - f(x^*)] + \frac{1}{2}\|x^0 - x^*\|^2 + \frac{\delta}{2\lambda'}\frac{\alpha^{mK} - \alpha^K}{\alpha - 1}\|\nabla f(x^*)\|^2 \tag{33}$$

Here we get the theorem.

**Proof of Theorem 2:**

from Lemma 4,8 and 9, we have:

$$\mathbb{E}[(\eta + \delta)f(x^{t+1}) - f(x^*) + \delta'(f(x^{t+1}) - f(x^t))]$$

$$\le (\eta + \delta')\left\langle g(x^t) - \nabla f(x^t), x^t - x^{t+1} \right\rangle - \left(\frac{1 + \delta\lambda' - 2\gamma_k}{2}\right)\|x^{t+1} - x^*\|^2 + \frac{\delta}{2\lambda'}\|\nabla f(x^*)\|^2$$

$$+ \frac{1 + \delta L_s/\lambda' - \delta v_s}{2}\|x^t - x^*\|^2 - \left(\frac{1 - \eta L_s - \delta L_s}{2} + \delta'\frac{2 - 2\gamma_k - \eta L_s}{2\eta}\right)\|x^t - x^{t+1}\|^2 + \frac{\eta L}{2n}\left(1 - \frac{1}{\theta}\right)\|x^t - \varphi_t^i\|^2 \tag{34}$$

Let $\lambda' = \frac{1}{\kappa_s}$, $\alpha = 1 + \delta/\kappa_s - 2\gamma_k < 1 + \frac{1}{m}$. Multiplying (34) by $\alpha^t$, and summing over the epoch $t = mk$ to $t = m(k+1) - 1$ for some $k \in \mathbb{N}_0$, we have:

$$\sum_{k=ms}^{m(s+1)-1} \alpha^t \mathbb{E}[(\eta + \delta)f(x^{t+1}) - f(x^*) + \delta'(f(x^{t+1}) - f(x^t))]$$

$$\leq \frac{1}{2}\alpha^{mk}\|x^{mk} - x^*\|^2 - \frac{1}{2}\alpha^{m(k+1)}\|x^{m(k+1)} - x^*\|^2 + \frac{\delta\kappa_s}{2}\sum_{k=ms}^{m(s+1)-1}\alpha^t\|\nabla f(x^*)\|^2 \tag{35}$$

$$+ \sum_{k=ms}^{m(s+1)-1}\alpha^t\mathbb{E}\Big[\frac{\eta L}{2n}(1 - \frac{1}{\theta})\|x^t - \varphi_t^i\|^2 + (\eta + \delta')\big\langle g(x^t) - \nabla f(x^t), x^t - x^{t+1}\big\rangle$$

$$- \big(\frac{1 - \eta L_s - \delta L_s}{2} + \delta'\frac{2 - 2\gamma_k - \eta L_s}{2\eta}\big)\|x^t - x^{t+1}\|^2\Big]$$

Let $\delta < \kappa_s(2\gamma_k + \frac{1}{m})$, we have:

$$\alpha^t < \alpha^{m(k+1)} \leq \alpha^{mk}\lim_{m \to \infty}(1 + \frac{1}{m})^m = e\alpha^{mk}$$

where $e$ is Euler's number. Summing the inequality from epoch $k = 0$ to $k = K - 1$:

$$\sum_{k=0}^{K-1}\alpha^k\mathbb{E}[(\eta + \delta)f(x^{t+1}) - f(x^*) + \delta'(f(x^{t+1}) - f(x^t))]$$

$$\leq \frac{1}{2}\|x^t - x^*\|^2 - \frac{1}{2}\alpha^{mK}\|x^{t+1} - x^*\|^2 + \frac{\delta\kappa_s}{2}\sum_{k=0}^{m(K-1)}\alpha^k s\|\nabla f(x^*)\|_\infty^2 \tag{36}$$

$$+ \sum_{k=0}^{K-1}\alpha^{mk}\sum_{t=mk}^{m(s+1)-1}e\mathbb{E}\Big[\frac{\eta L}{2n}(1 - \frac{1}{\theta})\|x^t - \varphi_t^i\|^2 + (\eta + \delta')\big\langle g(x^t) - \nabla f(x^t), x^t - x^{t+1}\big\rangle$$

$$- \big(\frac{1 - \eta L_s - \delta L_s}{2} + \delta'\frac{2 - 2\gamma_k - \eta L_s}{2\eta}\big)\|x^t - x^{t+1}\|^2\Big]$$

We use Lemma 7 with $\sigma_k = \alpha^m(k+1)$ to bound the MSE. Recall $\rho = \min\{\rho_M, \rho_F\}$ and $\delta\kappa_s - 2\gamma_k \leq \frac{\rho}{2}$. This choice for $\sigma_s$ satisfies the conditions of Lemma 7 because $\alpha^{mk}(1 - \rho)^{mk} \leq \alpha^{m(k-1)}(1 - \rho/2)^{mk}$ We use the fact that the gradient oracle is memory-biased to bound the term $\frac{1}{n}\sum_{i=1}^n\|x^t - \varphi_k^i\|^2$ and $\big\langle g(x^t) - \nabla f(x^t), x^t - x^{t+1}\big\rangle \leq \|g(x^t) - \nabla f(x^t)\| \cdot \|x^t - x^{t+1}\|$. This leaves

$$\eta\sum_{k=0}^{K-1}\alpha^k\mathbb{E}[(1 + \delta)f(x^{t+1}) - f(x^*) + \delta'(f(x^{t+1}) - f(x^t))]$$

$$\leq \frac{1}{2}\|x^t - x^*\|^2 - \frac{1}{2}\alpha^{mK}\|x^{t+1} - x^*\|^2 + \frac{\delta\kappa_s}{2}\sum_{k=0}^{m(K-1)}\alpha^k\|\nabla f(x^*)\|^2 \tag{37}$$

$$+ C\sum_{k=0}^{K-1}\alpha^{mk}\sum_{t=mk}^{m(s+1)-1}e\mathbb{E}\|x^t - x^{t+1}\|^2$$

Where $C = e\Big(\frac{\eta L_s B_1}{2}(1 - \frac{1}{\theta}) + (\eta + \delta')\sqrt{2\Theta}L_s - \big(\frac{1 - \eta L_s - \delta L_s}{2} + \delta'\frac{2 - 2\gamma_k - \eta L_s}{2\eta}\big)\Big).\delta' = \frac{L_s^2}{v_s}\eta$. We see $C$ is zero if:

$$\delta = \frac{2\frac{L_s^2}{v_s} + 1 - (L_s B_1(1 - \frac{1}{\theta}) + (\frac{L_s^2}{v_s} + 1)L_s(2\sqrt{2\Theta} + 1))\eta - 2\gamma_k}{L_s}$$

Recalling that we have $\frac{1 + 2n\gamma_k}{n\kappa_s} \leq \delta \leq 2\frac{\gamma_k}{\kappa_s}$. So we have

$$\frac{1 + 2n\gamma_k}{n\kappa_s} \leq \frac{2\frac{L_s^2}{v_s} + 1 - (L_s B_1(1 - \frac{1}{\theta}) + (\frac{L_s^2}{v_s} + 1)L_s(2\sqrt{2\Theta} + 1))\eta - 2\gamma_k}{L_s} \leq 2\frac{\gamma_k}{\kappa_s}$$

That is:

$$\frac{\frac{2\gamma_k L_s}{\kappa_s} + 2\gamma_k - (2\frac{L_s^2}{v_s} + 1)}{L_s B_1(1 - \frac{1}{\theta}) + (\frac{L_s^2}{v_s} + 1)L_s(2\sqrt{2\Theta} + 1)} \leq \eta \leq \frac{\frac{L_s \kappa_s + 2n\gamma_k L_s}{n\kappa_s} + 2\gamma_k - (2\frac{L_s^2}{v_s} + 1)}{L_s B_1(1 - \frac{1}{\theta}) + (\frac{L_s^2}{v_s} + 1)L_s(2\sqrt{2\Theta} + 1)}$$

So the step size in the theorem statement ensures $C = 0$ we have

$$\eta(1 + \delta + \delta') \sum_{k=0}^{K-1} \alpha^k \mathbb{E}[f(x^{t+1}) - f(x^*)] - \delta' \sum_{k=0}^{K-1} \alpha^k \mathbb{E}[f(x^t) - f(x^*)]$$

$$\leq \frac{1 + \delta L_s/\lambda' - \delta v_s}{2}\|x^t - x^*\|^2 - \frac{1 + \delta L_s/\lambda' - \delta v_s}{2}\alpha^{mK}\|x^{t+1} - x^*\|^2 + \frac{\delta}{2\lambda'} \sum_{k=0}^{m(K-1)} \alpha^k \|\nabla f(x^*)\|^2$$

$$(38)$$

Since $\delta' = \frac{L_s}{v_s}$ and $\eta < \frac{1}{L_s}$, we would like to show that $(1 + \delta + \delta') \geq \alpha\delta'$ so that the terms on the first line telescope.

$$\eta\delta'\alpha^{mK}\mathbb{E}[f(x^{mK}) - f(x^*)] + \frac{1}{2}\alpha^{mK}\|x^{mK} - x^*\|^2$$

$$\leq \eta\delta'\mathbb{E}[f(x^0) - f(x^*)] + \frac{1}{2}\|x^0 - x^*\|^2 + \frac{\delta}{2\lambda'}\frac{\alpha^{mK} - \alpha^K}{\alpha - 1}\|\nabla f(x^*)\|^2$$

$$(39)$$

Here we get the theorem.

# D PROOF FOR ZEROTH-ORDER ALGORITHM

## D.1 GRADIENT ORACLE IN ZEROTH-ORDER OPTIMIZATION

we define:

$$\hat{\nabla}_F f(x, u, \mu) = \frac{\sum_{i=1}^n f_i(x + \mu u) - f_i(x)}{\mu} u_F$$

To be convenience, we define:

$$\hat{\nabla}_F f_i(x, u) = \hat{\nabla}_F f_i(x, u, \mu); \hat{\nabla}_F f(x, u) := \hat{\nabla}_F f(x, u, \mu)$$

First we need to get the algorithm. We have two vision, first is vr and the second is dvr. Here is the dvr:

**Proof:**

$$\mathbb{E}\|v - x^*\|^2 \leq \mathbb{E}\|x^t - x^*\|^2 + \eta^2 \mathbb{E}\|\hat{g}_{\mathcal{I}}^t(x^t)\|_2^2 - 2\eta \left\langle x^t - x^*, \mathbb{E}\hat{g}_{\mathcal{I}}^t(x^t) \right\rangle$$

$$\leq \mathbb{E}\|x^t - x^*\|^2 + \eta^2 \mathbb{E}\|\hat{g}_{\mathcal{I}}^t(x^t)\|_2^2 - 2\eta \left\langle x^t - x^*, \frac{1}{\theta}\sqrt{\frac{s}{d}}\nabla_F f\left(x^t\right) + \sqrt{\frac{s}{d}}(1 - \frac{1}{\theta})\nabla_F f\left(w^t\right) + \frac{L\mu}{2}\tau_{i,t} \right\rangle$$

$$\leq \mathbb{E}\|x^t - x^*\|^2 + \eta^2 \mathbb{E}\|\hat{g}_{\mathcal{I}}^t(x^t)\|_2^2 - \frac{2\eta}{\theta}\sqrt{\frac{s}{d}}\left[f(x^t) - f(x^*)\right] - 2\eta(1 - \frac{1}{\theta})\sqrt{\frac{s}{d}}\left[f(x^0) - f(x^*)\right]$$

$$- (\frac{v_s\eta}{\theta})\sqrt{\frac{s}{d}}\|x^t - x^*\|_2^2 - \eta v_s(1 - \frac{1}{\theta})\sqrt{\frac{s}{d}}\|x^0 - x^*\|_2^2 - 2\eta(1 - \frac{1}{\theta})\sqrt{\frac{s}{d}}\left[f(x^t) - f(x^0)\right]$$

$$+ \eta L_s(1 - \frac{1}{\theta})\sqrt{\frac{s}{d}}\|x^0 - x^t\|_2^2 + \frac{\eta}{\lambda}\sqrt{\frac{s}{d}}\|\tau_{i,t}\|^2 + \sqrt{\frac{s}{d}}\lambda\eta\|x^t - x^*\|^2$$

$$\leq (1 - \frac{\eta v_s}{\theta}\sqrt{\frac{s}{d}} + \lambda\eta\sqrt{\frac{s}{d}})\|x^t - x^*\|_2^2 - \eta v_s(1 - \frac{1}{\theta})\sqrt{\frac{s}{d}}\|x^0 - x^*\|_2^2$$

$$+ \eta L_s(1 - \frac{1}{\theta})\sqrt{\frac{s}{d}}\|x^0 - x^t\|_2^2 - \frac{2\eta}{\theta}\sqrt{\frac{s}{d}}\left[f(x^t) - f(x^*)\right] - 2\eta(1 - \frac{1}{\theta})\sqrt{\frac{s}{d}}\left[f(x^0) - f(x^*)\right]$$

$$+ 2\eta L_s(1 - \frac{1}{\theta})\sqrt{\frac{s}{d}}\left[f(x^0) - f(x^t)\right] + \frac{\eta}{\lambda}\sqrt{\frac{s}{d}}\|\tau_{i,t}\|^2 + \eta^2 \mathbb{E}\|\hat{g}_{\mathcal{I}}^t(x^t)\|_2^2$$

$$\leq (1 - \frac{\eta v_s}{\theta}\sqrt{\frac{s}{d}} + \eta L_s 2(1 - \frac{1}{\theta})\sqrt{\frac{s}{d}} + \lambda\eta\sqrt{\frac{s}{d}})\|x^t - x^*\|_2^2 - (\eta v_s(1 - \frac{1}{\theta}) - 2\eta L_s(1 - \frac{1}{\theta}))\sqrt{\frac{s}{d}}\|x^0 - x^*\|_2^2$$

$$- 2\eta\sqrt{\frac{s}{d}}\left[f(x^t) - f(x^*)\right] + \frac{\eta}{\lambda}\sqrt{\frac{s}{d}}\|\tau_{i,t}\|^2 + \eta^2 \mathbb{E}\|\hat{g}_{\mathcal{I}}^t(x^t)\|_2^2$$

$$\tag{40}$$

For any $i \in [n]$ and $x$ with $supp(x) \subset \mathcal{I}$, consider:

$$\phi_i(x) = f_i(x) - f(x^*) - < \nabla f_i(x^*), x - x^* >$$

Since $\nabla\phi_i(x^*) = \nabla f_i(x^*) - \nabla f_i(x^*) = 0$, we have $\phi_i(x^*) = min_x\phi(x)$, which implies:

$$0 = \phi_i(x^*) \leq min_\eta\phi_i(x - \eta\nabla_F\phi_i(x)) \leq \min_\eta \phi_i(x) - \eta\|\nabla_F\|^2 + \frac{L_s\eta^2}{2}\|\nabla_F\phi_i(x)\|_2^2$$

$$= \phi_i(x) - \frac{1}{2L_s}\|\nabla_F\phi_i(x)\|_2^2$$

$$\tag{41}$$

where the second inequality follows from the RSS condition and the last equality follows from the fact that $\eta = \frac{1}{L_s}$ minimizes the function. From (41), we have:

$$\|\nabla_F f_i(x) - \nabla_{\mathcal{I}} f_i(x^*)\|_2^2 \leq 2L_s[f_i(x) - f_i(x^*) - < \nabla_F f_i(x^*), x - x^* >]. \tag{42}$$

Since the sampling of $i$ from $[n]$ is uniform, we have from (42)

$$\mathbb{E}\|\nabla_F f_i(x) - \nabla_{\mathcal{I}} f_i(x^*)\|_2^2 = \frac{1}{n}\|\nabla_F f_i(x) - \nabla_{\mathcal{I}} f_i(x^*)\|_2^2 \leq 2L_s[F(x) - F(x^*) - < \nabla_F F(x^*), x - x^* >]$$

$$\leq 2L_s[F(x) - F(x^*) + < \nabla_F F(x^*), x - x^* >] \leq 4L_s[F(x) - F(x^*)]$$

$$\tag{43}$$

where the last inequality is from the restricted convexity of $F(x)$ and the fact that $\|(x - x^*)_{\mathcal{I}}^C\|_0 = 0$

By the definition of $g$ in (3.4), we can verify the second claim as:

$$\mathbb{E}\|g(x)\|_2^2 = \mathbb{E}\|\frac{1}{\theta}\hat{\nabla}_{\mathcal{I}}f_i\left(x^t, u_k\right) - \frac{1}{\theta}\hat{\nabla}_{\mathcal{I}}f_i\left(w_k, u_k\right) + \hat{\nabla}_{\mathcal{I}}f\left(w_k, u\right)\|$$

$$\leq \frac{4}{\theta^2}\mathbb{E}\|u_{F,i}u^T\nabla_{\mathcal{I}}f_{i^t}(x) - u_{F,i}u_i^T\nabla_{\mathcal{I}}f_{i^t}(x^*)\|^2 + 4\mathbb{E}\|\nabla_F F(x^*) + \tau\|^2$$

$$+ 4\|u_F u^T(\nabla_{\mathcal{I}}f_i(\varphi) - \nabla_{\mathcal{I}}f_i(x^*)) - \frac{1}{\theta}u_{F,i}u_i^T(\nabla_{\mathcal{I}}f_i(\varphi) - \nabla_{\mathcal{I}}f_i(x^*))\|^2$$

$$+ \mathbb{E}\|u_F\|^2\|u\|^2\|(\nabla_{\mathcal{I}}f_i(\varphi) - \nabla_{\mathcal{I}}f_i(x^*)) - \mathbb{E}(\nabla_{\mathcal{I}}f_i(\varphi) - \nabla_{\mathcal{I}}f_i(x^*))\|^2$$

$$\leq \frac{4s}{\theta^2 d}\mathbb{E}\|\nabla_{\mathcal{I}}f_{i^t}(x) - \nabla_{\mathcal{I}}f_{i^t}(x^*)\|^2 + (8 + \frac{4}{\theta^2})\frac{q}{d}\|\nabla_{\mathcal{I}}f_{i^t}(\varphi) - \nabla_{\mathcal{I}}f_{i^t}(x^*)\|^2 \qquad (44)$$

$$+ 4\mathbb{E}\|\nabla_F F(x^*) + \tau\|^2$$

$$\leq (\frac{4s}{\theta^2 d}L_s - \frac{1}{2\eta}\sqrt{\frac{s}{d}})\mathbb{E}\|x - x^*\|^2 + (8 + \frac{4}{\theta^2})\frac{q}{d}L_s\|\varphi - x^*\|^2$$

$$+ \frac{2}{\eta}\sqrt{\frac{s}{d}}\left[f(x^t) - f(x^*)\right] + 4\mathbb{E}\|\nabla_F F(x^*) + \tau\|^2$$

So we have:

$$\mathbb{E}\|v - x^*\|^2 \leq (1 - \frac{\eta v_s}{\theta}\sqrt{\frac{s}{d}} + \eta L_s 2(1 - \frac{1}{\theta})\sqrt{\frac{s}{d}} + \lambda\eta\sqrt{\frac{s}{d}} + \eta^2((\frac{4s}{\theta^2 d}L_s - \frac{1}{2\eta})))\|x^t - x^*\|_2^2$$

$$- (\eta v_s(1 - \frac{1}{\theta})\sqrt{\frac{s}{d}} - (\eta L_s(1 + \frac{1}{\delta})(1 - \frac{1}{\theta}))\sqrt{\frac{s}{d}} - \eta^2 L_s\frac{4s}{\theta^2 d} - 8\eta^2 L_s\frac{q}{d})\|x^0 - x^*\|_2^2$$

$$+ \sqrt{\frac{s}{d}}\|\tau_{i,t}\|^2 + 4\eta^2\mathbb{E}\|\nabla_F F(x^*) + \tau_{i,t}\|^2$$

$$(45)$$

That is:

$$\mathbb{E}\|x^{t+1} - x^*\|^2 \leq \alpha(1 - \sqrt{\frac{s}{d}}(\frac{\eta v_s}{\theta} - \eta L_s 2(1 - \frac{1}{\theta}) - \lambda\eta) + \eta^2(\frac{4}{\theta^2}L_s - \frac{1}{2\eta}\|x^t - x^*\|_2^2$$

$$- \alpha(\eta v_s(1 - \frac{1}{\theta})\sqrt{\frac{s}{d}} - (\eta L_s(1 + \frac{1}{\delta})(1 - \frac{1}{\theta}))\sqrt{\frac{s}{d}} - \eta^2 L_s\frac{4s}{\theta^2 d} - 8\frac{q}{d}\eta^2 L_s)\|x^0 - x^*\|_2^2$$

$$+ \alpha\sqrt{\frac{s}{d}}\|\tau_{i,t}\|^2 + \alpha 4\eta^2\mathbb{E}\|\nabla_F F(x^*) + \tau_{i,t}\|^2$$

$$(46)$$

We let $\beta = \alpha(1 - \frac{\eta v_s}{\theta}\sqrt{\frac{s}{d}} + \eta L_s 2(1 - \frac{1}{\theta})\sqrt{\frac{s}{d}} + \lambda\eta\sqrt{\frac{s}{d}} + \eta^2\frac{4s}{\theta^2 d}L_s 2 - 2\eta$, and $\gamma = \alpha(\eta v_s(1 - \frac{1}{\theta})\sqrt{\frac{s}{d}} - 2\eta L_s(1 - \frac{1}{\theta})\sqrt{\frac{s}{d}} - \eta^2 L_s\frac{4s}{\theta^2 d} - 8\eta^2 L_s\frac{q}{d})$If we use a count $\theta$, Then we have:

$$\mathbb{E}\|x^m - x^*\|^2 \leq (\beta^m - \frac{\beta^m - 1}{\beta - 1}\gamma)\mathbb{E}\|x^0 - x^*\|_2^2$$

$$+ \frac{\beta^m - 1}{\beta - 1}\alpha\sqrt{\frac{s}{d}}\|\tau_{i,t}\|^2 + \frac{\beta^m - 1}{\beta - 1}\alpha 4\eta^2\mathbb{E}\|\nabla_F F(x^*) + \tau_{i,t}\|^2$$

$$(47)$$

# E   MORE ALGORITHM

# F   MORE EXPERIMENTS

**Ridge Regression**   Ridge regression is a commonly used biased estimation linear regression method in statistics and machine learning. It improves the stability and generalization ability of the model by adding a regularization term ($\ell_2$ norm) to the least squares method. For consistency in narration, we consider the expression for ridge regression as follows:

$$f_i(\omega) = (x_i^\top \omega - y_i)^2 + \frac{\lambda}{2}\|\omega\|_2^2,$$

---

**Algorithm 2** StochAstic Recursive grAdient algoritHm with Hard-Thresholding(SARAH-HT)

---

**Input:** Learning rate $\eta$, maximum number of iterations $T$, initial point $x^0$, SVRG update frequency $m$, and number of coordinates to keep at each iteration $k$.

**Output:** $x^T$.

    **for** $r = 1, \ldots, T$ **do**

        $x^{(0)} = x^{r-1}$;

        $v^{(0)} = \frac{1}{n} \sum_{i=1}^{n} \nabla f_i(x^{(0)})$;

        $x^{(1)} = x^{(0)} - \eta v^{(0)}$

        **for** $t = 0, 1, \ldots, m-1$ **do**

            Randomly sample $i_t \in \{1, 2, \ldots, n\}$;

            $v^{(t+1)} = \nabla f_{i_t}(x_{t+1}) - \nabla f_{i_t}(x_t + v^{(t)})$;

            $x^{(t+1)} = \mathcal{H}_k(\boldsymbol{x}^{(t)} - \eta v^{t+1})$;

        **end for**

        $x^r = x^{(t')}$, random $t' \in [m-1]$

    **end for**

---

**Algorithm 3** Stochastic bias variance reduced Hard-Thresholding algorithm (BVR-SHT)

---

**Input:** Learning rate $\eta$, maximum number of iterations $T$, initial point $x^0$, SVRG update frequency $m$, and number of coordinates to keep at each iteration $k$.

**Output:** $x^T$.

    **for** $r = 1, \ldots, T$ **do**

        $x^{(0)} = x^{r-1}$;

        Compute $\mu = \frac{1}{n} \sum_{i=1}^{n} \nabla f_i(x^{(0)})$;

        **for** $t = 0, 1, \ldots, m-1$ **do**

            Randomly sample $i_r \in \{1, 2, \ldots, n\}$;

            $\bar{x}^{(t+1)} = x^{(t)} - \eta(\frac{1}{\theta}(\hat{\nabla} f_{i_t}(x^{(t)}) - \hat{\nabla} f_{i_t}(x^{(0)})) + \mu)$;

            $x^{(t+1)} = \mathcal{H}_k(\bar{x}^{(t+1)})$;

        **end for**

        $x^r = x^{(t')}$, random $t' \in [m-1]$

    **end for**

---

where $\lambda$ is the regularization parameter, $\omega$ is the model weight. We randomly generate each $x_i$ from a hyper-sphere with a unit radius in $\mathbb{R}^d$, and the true model weight $\omega^*$ is drawn from a Gaussian distribution $\mathcal{N}(0, I_{d \times d})$. Each $y_i$ is calculated as $y_i = x_i^T \omega^*$. In our ZO comparative experiment, we set the constants as such: $n = 10, d = 5, \lambda = 0.5$. Before training, we preprocess each column by subtracting its mean and dividing it by its empirical standard deviation. We run each algorithm with $k = 3, q = 200, \mu = 10^{-4}, s_2 = d = 5$, and for the variance reduced algorithms, we choose $m = 10$ and bias coefficient $\theta = 2$. For all algorithms, the learning rate $\eta$ is found through grid-search in $\{0.005, 0.01, 0.05, 0.1, 0.5\}$. We choose the $\eta$ giving the lowest function value (averaged over several runs) at the end of training. We stop each algorithm once its IZO reaches 80,000. All curves are averaged over 3 runs, and we plot their mean and standard deviation in Figure 3. It can be observed that BVR-SZHT converges faster than other algorithms and reaches lower loss values.

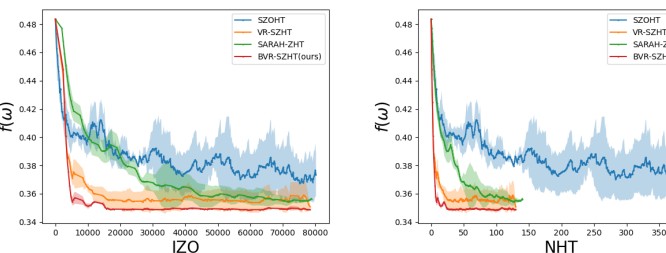

Figure 2: Loss values of ZO algorithms in ridge regression tasks

In the first-order part, we define bias coefficient $\theta = 2$ or $n$ and use gradients instead of zeroth-order oracle. All curves are also averaged over 3 runs, and we plot their mean and standard deviation in Figure 2. It can be observed that SARAH-HT converges faster than other algorithms.

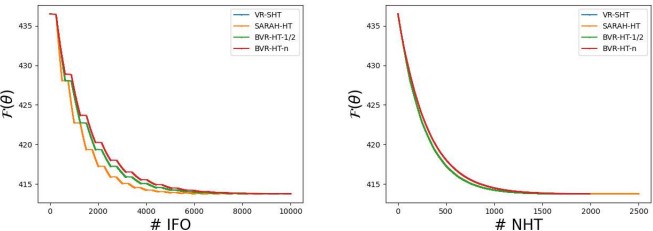

Figure 3: Loss values of FO algorithms in ridge regression tasks

**Sensitivity analysis** To validate the bias cancellation effect of SARAH in the first-order hard-thresholding algorithm and BSVRG-HT in the zeroth-order hard-thresholding algorithm, we conducted sensitivity analysis based on ridge regression experiments. In first-order algorithms, since the bias from hard thresholding is restricted solely by $k$, we subtracted the loss function of BVR-SHT from that of SARAH-HT (as shown in Figure 3). Due to the inevitable oscillations in the early stages of convergence, which can affect observation, we focus more on the stable phase of the iterations. As $k$ increases, the difference in loss functions grows, indicating that SARAH shows a greater advantage over BVRSZHT when variance is large, thanks to its stronger variance cancellation effect. For the zeroth-order algorithm, we conduct a sensitivity analysis on $\mu$ based on the ridge regression experiments for BVR-SZHT and SARAH. We emphasize once again that $\mu$ can control the bias of the zeroth-order gradient. We observed that BVR-SZHT is not sensitive to changes in $\mu$, whereas SARAH's convergence gradually worsens as $\mu$ increases. This demonstrates that BVR-SZHT can partially offset the bias introduced by the zeroth-order gradient.

**Black-box Adversarial Attacks** Adversarial attacks trick machine learning models by adding carefully designed subtle perturbations to inputs, leading to mispredictions. Black-box adversarial

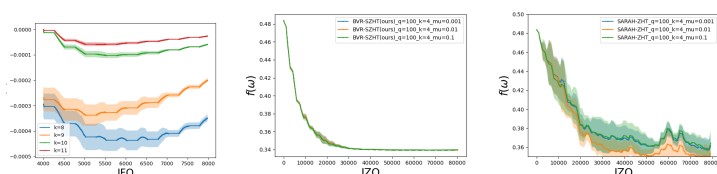

Figure 4: Sensitive Analysis for FO(left) and ZO(right)

attacks occur when attackers can't access a model's internals and must deduce its behavior from inputs and outputs. The Black-box attack method is closer to real-world attack scenarios. Therefore, we consider a few-pixel universal adversarial attack scenarios and assume there is a well-trained classifier that can only be accessed as a black box. In this scenario, zeroth-order algorithms excel over first-order ones in black-box settings as they don't need model gradients, estimating them through output queries instead. As is usual in black-box adversarial attacks, we maximize the following Carlini-Wagner loss (Carlini & Wagner, 2017; Chen et al., 2017), which promotes the model the model to make incorrect predictions:

$$f_i(\omega) = \max\{F_{y_i}(\text{clip}(x_i + \omega)) - \max_{j \neq y_i} F_j(\text{clip}(x_i + \omega)), 0\},$$

where $F$ denotes a pre-trained model, $x_i$ is the $i$-th image (rescaled to have values in $[-0.5, 0.5]$) with true class $y_i$, clip denotes the clipping operation into $[-0.5, 0.5]$, $\omega$ is the universal perturbation that we seek to optimize, and each $F_j$ outputs the log-probability of image $x_i$ being of class $n$ as predicted by the model ($j \in \{1, .., J\}$, $J$ is the number of classes, similarly to (Chen et al., 2017; Huang et al., 2019)). We use the pre-trained model on the CIFAR-10 as the model $F$. It can be obtained from the supplementary material of (de Vazelhes et al., 2022). Similarly to Liu et al. (2018), we evaluate the algorithms on a dataset of $n = 10$ images from the test-set of the CIFAR-10 dataset(Krizhevsky & Hinton, 2009). We set $k = 60$, $\mu = 0.001$, $q = 10$, $s_2 = d = 3,072$, the number of inner iterations of the variance reduced algorithms to $m = 10$ and the bias coefficient $\frac{1}{\theta} = 0.65$. We check at each iteration the number of IZO, and we stop training if it exceeds 600. Finally, we grid-search the learning rate $\eta$ in $\{0.001, 0.005, 0.01, 0.05\}$ and select the one that minimizes the loss value for each algorithm. The training curves are presented in Figure 5. We can observe that BVR-SZHT achieved the lowest loss value and showed significant performance improvement compared to VR-SZHT in this tasks.

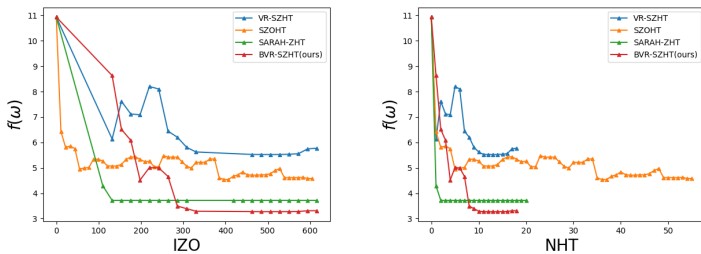

Figure 5: Loss values of ZO algorithms in black-box adversarial attack

**Sparse Feature Selection**    Feature selection is a crucial step in reducing dimensionality and improving model interpretability, especially when dealing with high-dimensional biological datasets like scRNA-seq data. In our work, we applied several feature selection algorithms, BSVRG-HT, SAGA, and SARAH-HT, to efficiently select a subset of features that best represent the underlying biological signals. SAGA-LASSO, a popular approach for sparse logistic regression, uses the L1 penalty to encourage sparsity while leveraging stochastic optimization to solve large-scale problems efficiently. We conducted feature selection on scRNA-seq data and MINST/CIFAR-10 datasets from colorectal cancer cell lines. Following feature selection, we trained a deep neural network (DNN) to classify cell types based on the selected features. We optimized the hyperparameters, such as

learning rates and batch sizes, for each feature selection algorithm to maximize the classification accuracy. The results of our experiments demonstrate the effectiveness of these methods in high-dimensional biological settings. BVRSZHT and SARAH both provided significant performance improvements in feature reduction while maintaining high accuracy. The selected features were subsequently used to train the DNN classifier, resulting in robust and interpretable predictions of cell type identities.

| Dataset | Algorithm | Accuracy | Num_Features | Selection_Time (s) |
|---------|-----------|----------|--------------|--------------------|
| Cancer | BVRSZHTn | 0.8850 | 2863 | 71.92 |
| Cancer | SAGA-LASSO | 0.9204 | 3470 | 645.07 |
| Cancer | BVRSZHT12 | 0.8673 | 2863 | 68.30 |
| Cancer | VRSZHT | 0.8496 | 2863 | 73.12 |
| Cancer | SARAH | 0.8938 | 2863 | 65.66 |
| CIFAR-10 | BVRSZHTn | 0.4575 | 1843 | 153.32 |
| CIFAR-10 | SAGA-LASSO | 0.5102 | 3053 | 5148.21 |
| CIFAR-10 | BVRSZHT12 | 0.5109 | 1843 | 152.18 |
| CIFAR-10 | VRSZHT | 0.5029 | 1843 | 150.75 |
| CIFAR-10 | SARAH | 0.5126 | 1843 | 153.08 |
| MNIST | BVRSZHTn | 0.9593 | 235 | 70.09 |
| MNIST | SAGA-LASSO | 0.9729 | 644 | 1131.67 |
| MNIST | BVRSZHT12 | 0.9563 | 235 | 70.43 |
| MNIST | VRSZHT | 0.9407 | 235 | 70.63 |
| MNIST | SARAH | 0.9616 | 235 | 64.00 |

Table 2: Reasult in sparse feature selesction

