# OpenReview forum: "Fight Fire with Fire: Multi-biased Interactions in Hard-Thresholding"
_ICLR.cc/2025/Conference — Submitted to ICLR 2025_

### Official Review · Reviewer_PAJk · 2024-10-25

**Soundness:** 2
**Presentation:** 1
**Contribution:** 2
**Rating:** 3
**Confidence:** 4

**Summary:**

This paper revisit the classic Hard-Thresholding algorithm from the viewpoint of biased gradients. The biases are categorized into "memory-biased" and "recursively-biased" ones. The performance of these two categories are respectively examined with First-order and Zeroth-order Hard-Thresholding algorithms. Experiments including black-box adversarial attacks and ridge regressions are conducted to validate the empirical advantages of the proposed approaches, and analyze the parameter sensitivity. However, the writing and presentation of this submission are very poor, making the paper extremely difficult to interpret and evaluate. Consequently, I recommend a complete rewrite of the paper.

**Strengths:**

The paper is based on solid theoretical analysis.

**Weaknesses:**

The major concern for this paper is its writing and presentation, which serves as the primary hurdle to understand the paper. Below are my point-by-point comments.

1. The paper is not self-contained -- a lot of notations come out of the blue without any definition. As a consequence, the readers of this paper have to search for related works to understand them. Unfortunately, there is even no reference for some notations. For example, symbols $\varphi$, $\nu$, and $\mathbb{N}_0$ appear in line 140, while there is not any definition/explanation/reference regarding their meaning.

2. The notations are inconsistently used throughout the paper. For instance, in Assumption 3, the smoothness constant is denoted as $L_s$, but the function is referred to as $\rho_s^+$-strongly smooth. Additionally, $v_s$ and $L_s$ are respectively used for RSC and RSS constants in Assumptions 2 and 3, while in Section 4 the function $f_i$ is assumed $v_s$-RSS and $L_s$-RSC (rather than $v_s$-RSC and $L_s$-RSS). Other examples include Lemma 1, where $\gamma_k$ appears in the equation, but line 203 uses $\gamma$ without a subscript.

3. Many sentences lack clear logic. Remark 1 claims that "the MSE of $g(x)$ does not completely determine the convergence", but the next sentence still suggests "using the MSE of $\nabla_{HT}$ as a substitute". Furthermore, Remark 4 states that "the
Hard-Threshold can counteract some bias, thus accelerating convergence", which directly conflicts with Remark 1. Similar logical inconsistencies appear in e.g., Section 3.3. Additionally, Sections 3 and 4 contain many lemmas and theorems that are either poorly explained or entirely unexplained; cf. Theorem 3.

4. The experiments are also vague, and there is no reference provided in Section 5. The introduction mentions "large-scale machine learning". In contrast, the numerical tests are limited to merely ridge regression in a space of dimensionality $5$.

5. There are also numerous grammatical error, typos, and citations of improper formats. For instance, the phrase "recursively bias effectively counteracts some of the issues" appears multiple times, where "recursively" should be corrected to "recursive."

**Questions:**

I would suggest completely rewriting the paper.

---

> ### Author Response · Authors · 2024-11-23
>
> Thank you for your thorough review and detailed comments. We sincerely apologize for the poor writing and presentation of our submission, which clearly made it difficult to interpret and evaluate our work. We highly value your feedback and have worked to address the major concerns you raised. Below, we provide responses to each of your comments:
>
> ## 1. On Writing and Presentation
> We acknowledge that the quality of writing and presentation in the current version is insufficient. We apologize for any inconvenience this has caused. We have already begun revising the paper to ensure clearer definitions, consistent notation, and improved logical flow. In our next version, we will provide more detailed explanations of all notations, include necessary references, and refine the writing to eliminate grammatical errors and typos. Your specific examples of unclear notations and inconsistencies are extremely helpful, and we will address them systematically in the revised manuscript.
>
> ## 2. On Logical Inconsistencies (Point 3)
> Regarding your concern about logical inconsistencies, we believe there is a misunderstanding. Specifically, in Remark 4, we state that "the **recursive biased algorithm for the Hard-Threshold** can counteract some bias, thus accelerating convergence," which refers to our proposed method. This does not suggest that the general hard-thresholding operation itself eliminates bias, as you implied.
>
> ## 3. On the Number of Experiments (Point 4)
> We respectfully disagree with your claim that our experiments are vague or insufficient. As summarized in your review, our submission includes two experiments: **black-box adversarial attacks** and **ridge regression**. This demonstrates the applicability of our approach to different tasks. While the numerical example in ridge regression is limited in dimensionality, the black-box adversarial attack experiment explicitly deals with large-scale data, aligning with the introduction's mention of "large-scale machine learning." In addition, we have also conducted experiments on sparse feature selection.
>
>
> ## 5. On Grammatical Errors and Typos
> Thank you for pointing out specific issues such as the misuse of "recursively." We will correct these issues and perform a thorough proofreading of the paper to ensure proper grammar and consistent terminology.
>
> ---
>
> We deeply appreciate your feedback, which has highlighted key areas for improvement in our submission. By addressing these issues, we aim to make our contributions more accessible and convincing in the next version. Thank you again for your insightful comments and suggestions.

---

> > ### Comment · Reviewer_PAJk · 2024-12-02
> >
> > I have checked the updated paper, which addresses some of my concerns. However, there remains remarkable room for improvement in the writing, presentation of theoretical findings, and "large-scale machine learning" experiments. Consequently, I will maintain my initial score.

---

### Official Review · Reviewer_Gwhn · 2024-11-01

**Soundness:** 1
**Presentation:** 1
**Contribution:** 2
**Rating:** 3
**Confidence:** 3

**Summary:**

The paper studies biases in gradient descent and how they might cancel issues caused by hard thresholding. The result of this study are two new algorithms for hard-thresholding stochastic gradient descent.

**Strengths:**

I think this paper studied a topic that is not very explored. It might give new inside and new ideas.

The success of lasso and l1 regularization cast a shadow on l0 regularization, and I don't remember seeing a lot about it in the literature. Studying it might give new insight and propel new ideas.

**Weaknesses:**

I believe the papers could improve in their presentation, clarity, and empirical validation.

**Numerical experiments**:  I feel the authors don't provide a very comprehensive set of experiments mentioned in the main text
   - It is only one experiment with ridge regression in the main text (and it uses simulated data only). There seem to be additional experiments in the appendix, but they are not even mentioned in the main text
  - I think the algorithm should be directly compared with LASSO, since it is a very popular to promote sparsity. It could be interesting to directly compare convergence and runtime to for instance SAGA implemented on Lasso.

**Presentation and style**: I would recommend significant improvements in presentation and style
  - I feel the results are presented with very little discussion. I feel the main part of the paper is just a collection of results. Could you maybe provide some interpretation for theorems 1, 2, and 3.
   - There are a lot of words that are Capitalized without a clear reason. For instance, Hard-Threshold, Zeroth-Order,  First-Order.
   - I would recommend improving the use of use of natbib. Avoid writing the name and then using citep, i.e. line 060, "William (de Vazelhes et al , 2022)", or in line 063 "Yuan (Yuan et al, 2024)". Maybe just use citet instead
  - The use of the notation $\nabla^{BSAGA}_t$ feels a bit unusual, and a bit confusing with the notation used for gradient
  - The figures have a hard-to-read title, captions, and numbering.
  - The excessive use of abbreviations makes the text hard to read.

**Questions:**

About the numerical setup
- It is a bit unclear to me why to use a  l_0 regularization together with ridge regression. One argument in the introduction is that l0 regularization avoids hyperparameters, but in the example you re-introduce hyperparameters. How the algorithm performs in the estimation with this constraint?

Other question:
- What are the asymptotic bounds on the computational complexity of iteration of the algorithm with hard thresholding?

---

> ### Author Response · Authors · 2024-11-23
>
> Thank you for your insightful review. We sincerely hope that the main concerns raised in the review can be clarified by the following responses.
>
> ---
> ### On the Difference Between $ l_0 $ Optimization and $ l_1 $ Optimization
> First, we are pleased to discuss the differences between $ l_0 $ optimization and $ l_1 $ optimization. Indeed, these two methods have better applications in different fields. It is undeniable that $ l_1 $ optimization typically achieves better accuracy, which has made it the focus of more widespread research. However, for many scenarios with strict sparsity requirements, $ l_0 $ optimization remains an important algorithm. Compared to $ l_1 $ optimization, $ l_0 $ optimization naturally has lower computational costs, making $ l_0 $-based algorithms faster in general. Additionally, in scenarios requiring strict sparsity, LASSO often struggles because it is difficult to directly specify the sparsity level. A typical example is sparse adversarial attacks, where the number of noise points must be limited, but LASSO cannot precisely control the sparsity. Similar issues arise in sparse feature selection tasks.\
> In addition, in our sparse feature selection experiment, we can see that compared to $l_1$ optimization, $l_0$ optimization is similar in accuracy but has smaller sparsity and faster computation time.
>
> ---
>
> ### On Numerical Experiments
>
> Thank you for your comments. We have included LASSO-SAGA in the experiments for first-order algorithms. Additionally, we have incorporated the adversarial attack experiment and the sparse feature selection experiment into the main text.
>
> ---
>
> ### On Presentation Style
>
> We have made the necessary adjustments to the presentation style. Thank you for pointing this out.
>
> ---
>
> > It is a bit unclear to me why to use $ l_0 $ regularization together with ridge regression. One argument in the introduction is that $ l_0 $ regularization avoids hyperparameters, but in the example you re-introduce hyperparameters. How does the algorithm perform in estimation with this constraint?
>
> First, we would like to clarify that $ l_0 $ optimization is not solely about avoiding hyperparameters but about avoiding the computation of hyperparameters (compared to ISTA algorithms). When combining $ l_0 $-norm with $ l_2 $-norm, the model’s robustness to noise can be significantly enhanced. This is because sparse optimization, due to the omission of some features, is often susceptible to noise. In fact, using $ l_2 $-norm in sparse optimization problems is a common practice, not only in $ l_0 $ optimization but also in $ l_1 $ optimization, as seen in approaches like Elastic Net.
>
> ---
>
> > What are the asymptotic bounds on the computational complexity of iteration of the algorithm with hard thresholding?
>
> We note that if $ f $ has a $ k^* $-sparse unconstrained minimizer, which could happen in sparse reconstruction, or with overparameterized deep networks (see for instance [1] Assumption (2)]), then we would have $ \|\nabla f(x^*)\| = 0 $, and hence that part of the system error would vanish.*
>
> [1] Alexandra Peste, Eugenia Iofinova, Adrian Vladu, and Dan Alistarh. Ac/dc: Alternating compressed/decompressed training of deep neural networks. *Advances in Neural Information Processing Systems*, 34, 2021.

---

> > ### Comment · Reviewer_Gwhn · 2024-11-28
> >
> > I read the author's response and I thank them for the additional experiments. I keep my overall assessment of the paper

---

### Official Review · Reviewer_9ExX · 2024-11-02

**Soundness:** 2
**Presentation:** 2
**Contribution:** 1
**Rating:** 3
**Confidence:** 5

**Summary:**

This paper studies the performance of memory-biased gradient and recursively-biased gradient oracle in first-order and zeroth-order stochastic algorithms for L0-norm constrained optimization problems. Moreover, this paper also proposes several hard-thresholding algorithms of first-order algorithms (such as SARAH, BSVRG and BSAGA) and zeroth-order algorithms such as BVR-SZHT. Some experimental results are reported.

**Strengths:**

The paper is complete in format.

**Weaknesses:**

The one main contribution of this paper is to propose several first-order and zeroth-order hard-thresholding algorithms such as SARAH-HT, BSVRG-HT, and BSAGA-HT. There are much first-order and zeroth-order stochastic hard-thresholding algorithms. Therefore, the novelty of this paper is very limited.

**Questions:**

1.	What’s the advantage of the proposed first-order and zeroth-order stochastic hard-thresholding algorithms against related algorithms in terms of convergence rates and complexities?
2.	The experimental results are not convincing. The authors should compare the proposed algorithms with more recently proposed algorithms.
3.	Both the English language and equations in this paper need to be improved. For example,  Line 225: ‘In Hard-Thresholding algorirthm, The commonly used Zeroth-Order estimation is’.

---

> ### Author Response · Authors · 2024-11-23
>
> Thank you for your insightful review. We sincerely hope that the main concerns raised in your review can be clarified by the following responses.
>
> ---
>
> > The one main contribution of this paper is to propose several first-order and zeroth-order hard-thresholding algorithms such as SARAH-HT, BSVRG-HT, and BSAGA-HT. There are much first-order and zeroth-order stochastic hard-thresholding algorithms. Therefore, the novelty of this paper is very limited.
>
> While we acknowledge that there are existing studies on first-order hard-thresholding algorithms, we would like to highlight the following points regarding the novelty of our work:
> 1. **Limited Research on Zeroth-Order Hard-Thresholding**: To the best of our knowledge, only two prior works (William, 2023 and Yuan, 2024) have studied zeroth-order hard-thresholding algorithms.
>    - Our proposed algorithm framework, BVR-SZHT, generalizes existing zeroth-order hard-thresholding methods, including  VR-SZHT (2024) as a special case when $\theta=1$. This unified framework allows us to better analyze the role of gradient bias and demonstrate how appropriate parameter choices (e.g., $\theta>1$) lead to faster convergence.
> 2. **Gradient Bias Analysis**: Our work is the first to investigate the impact of gradient bias in both first-order and zeroth-order settings. This includes analyzing how memory and recursive biases interact with the hard-thresholding operator and using this understanding to design algorithms that improve convergence and efficiency.
>
> ---
>
> > 1. What’s the advantage of the proposed first-order and zeroth-order stochastic hard-thresholding algorithms against related algorithms in terms of convergence rates and complexities?
> ---
>
> -First-Order Algorithms:\
> Our BSVRG-HT algorithm encompasses SVRG-HT [1] as a special case. Specifically, SVRG-HT can be viewed as a particular instance of BSVRG-HT when $\theta$. This relationship is directly reflected in Theorem 1, where the advantages of BSVRG-HT in terms of improved convergence rates are clearly demonstrated. \
> -Zeroth-Order Algorithms:\
> Similarly, our BVR-SZHT algorithm generalizes VR-SZHT. By introducing adjustable parameters, BVR-SZHT effectively mitigates gradient instability issues inherent in VR-SZHT, resulting in faster convergence.\
> To support these claims, we have added a remark in the manuscript with detailed comparisons of convergence rates and complexities.
> > 2. The experimental results are not convincing. The authors should compare the proposed algorithms with more recently proposed algorithms.
>
> Thank you for your suggestion. We would like to clarify the following regarding the experimental comparisons:
> Zeroth-Order Algorithms:
> For zeroth-order hard-thresholding, we compared our proposed BVR-SZHT with the two most recent state-of-the-art algorithms:
> 1.VR-SZHT (2024): Included as a special case within our framework (when $\theta=1$).
> 2.SZOHT (2023): Another prominent recent work in this domain.
> Our experiments (e.g., few-point adversarial attack on CIFAR-10) consistently demonstrate that BVR-SZHT outperforms both VR-SZHT and SZOHT  in terms of convergence speed and computational efficiency. This validates the effectiveness of our approach.
> First-Order Algorithms:
> For first-order hard-thresholding, we compared SARAH-HT and BSVRG-HT with:
> 1.SVRG-HT: A recently proposed variant of the SVRG algorithm combined with hard-thresholding.
> 2.SAGA-LASSO: A widely recognized first-order method integrating SAGA with LASSO regularization.
> We recognize that the experimental section may not have provided sufficient details about the algorithms we compared. To address this, we have added necessary descriptions of all baseline algorithms to the revised paper, ensuring clarity and transparency. If there are any specific algorithms you believe must be included for comparison, we kindly ask for your suggestions. We are more than willing to conduct additional experiments to incorporate these comparisons.
>
> ---
>
> **> 3. Both the English language and equations in this paper need to be improved. For example, Line 225: ‘In Hard-Thresholding algorirthm, The commonly used Zeroth-Order estimation is’.**
>
> Thank you for pointing this out. We have carefully revised the manuscript. Overall language has been improved to ensure readability and flow.
> We sincerely hope these changes address your concerns and demonstrate the significance of our contributions.
> [1]Li, Xingguo, et al. "Nonconvex sparse learning via stochastic optimization with progressive variance reduction." arXiv preprint arXiv:1605.02711 (2016).

---

### Official Review · Reviewer_Sddm · 2024-11-04

**Soundness:** 3
**Presentation:** 3
**Contribution:** 3
**Rating:** 6
**Confidence:** 3

**Summary:**

The paper investigates how different types of biases interact in Hard-Thresholding (HT) optimization algorithms for L0-constrained problems. The authors reinterpret HT algorithms as biased gradient methods and explore two main types of biases: memory-biased and recursively-biased. They analyze how these biases interact both in First-Order (FO) and Zeroth-Order (ZO) optimization settings. Based on their theoretical analysis, they propose two new algorithms: SARAHT for FO optimization and BVR-SZHT for ZO optimization. The key insight is that recursive bias can help counteract HT-induced bias in FO settings, while memory bias works better with ZO estimation bias. They validate their findings through ridge regression experiments and black-box adversarial attacks.

**Strengths:**

1. The theoretical framework is novel with clean reinterpretation of HT algorithms as biased gradient methods and thorough analysis of bias interactions through bounded MSE conditions.
2. The technical contributions are sufficient. For example, the authors provide rigorous analysis on bias cancellation effects and detailed convergence analysis for both FO and ZO settings. The authors also provide clear characterization of how different biases interact under HT operators.
3. There are multiple metrics for evaluation such as IFO, IZO and NHT.

**Weaknesses:**

1. The experiments are somewhat weak.

- The main paper only presents ridge regression experiments, while important black-box adversarial experiments are deferred to the appendix.
- I recommend moving key adversarial attack results to the main paper, particularly those demonstrating the practical benefits of bias cancellation in zeroth-order optimization
- No evaluation on real-world large-scale datasets. Specifically, I recommend testing on: a) Sparse feature selection problems using MNIST/CIFAR-10 for computer vision, b) Gene expression datasets like Colon Cancer or Leukemia for bioinformatics applications, c) Text classification with sparse word embeddings using Reuters or 20 Newsgroups datasets. These datasets would demonstrate the practical utility of the proposed methods across diverse domains.

2. As for the theretical analysis, the discussion of when these assumptions might fail is limited and there is no analysis of what happens when conditions are violated.

- The Restricted Strong Convexity (RSC) and Restricted Strong Smoothness (RSS) assumptions are quite strong and their limitations should be discussed. Specifically, these assumptions may fail in:
a) Deep neural network optimization where loss landscapes are highly non-convex.
b) Problems with heavy-tailed noise where smoothness is violated.
c) High-dimensional settings where restricted eigenvalue conditions break down.

- The paper should analyze algorithm behavior when these conditions are violated and propose potential modifications or relaxations of the assumptions.

**Questions:**

1. How do the proposed algorithms perform on non-smooth optimization problems or problems where RSC/RSS conditions are violated?
2. What is the computational overhead of tracking and managing different types of biases compared to simpler approaches?
3. How would the bias interaction analysis extend to other sparsification operators beyond hard thresholding (e.g., soft thresholding)?
4. Can the framework be extended to handle constrained optimization problems while maintaining the bias cancellation benefits?

---

> ### Author Response · Authors · 2024-11-23
>
> We extend our sincere gratitude for dedicating your valuable time to reviewing our paper and for expressing your appreciation and support for our work. In the following, we will provide a comprehensive response to your review comments.
>
> ---
>
> > The experiments are somewhat weak.
>
> Thank you for your valuable suggestion, which provides us with great insights. We have supplemented our experiments with sparse feature selection tasks on the CIFAR-10, MNIST datasets, and a colon cancer dataset. These experiments highlight the performance of our proposed methods under practical settings. Additionally, we are conducting further experiments, and if time permits, we will provide additional results before the rebuttal period ends.
>
> ---
>
> > As for the theoretical analysis, the discussion of when these assumptions might fail is limited and there is no analysis of what happens when conditions are violated.
>
> We appreciate your insightful observation regarding the assumptions. It is important to clarify that the RSC (Restricted Strong Convexity) and RSS (Restricted Strong Smoothness) conditions are specifically designed for high-dimensional and non-convex problems. These conditions do not require global convexity or smoothness but instead impose curvature requirements along specific sparse directions ($\|x-y\|_0 < s$).
>
> Thus, concerns about points (a) and (c) are alleviated by the very nature of these assumptions, which are widely used in the field of sparse optimization[1,2]. Indeed, most work in this domain relies on these conditions.
>
> Regarding point (b), the impact of noise on hard-thresholding algorithms is an intriguing topic, but it is challenging to analyze it purely from the perspective of bias cancellation. This presents a valuable avenue for future work, which we intend to explore.
>
> ---
>
> > What is the computational overhead of tracking and managing different types of biases compared to simpler approaches?
>
> The computational cost of our framework is inherently low due to its structural simplicity. Specifically, BSAGA-HT and BSVRG-HT algorithm only modifies a single parameter compared to unbiased methods such as SAGA-HT and SVRG-HT. This allows our approach to achieve faster convergence without introducing additional computational overhead during iterations.
>
> Preliminary experimental results further support this claim, demonstrating that our method achieves convergence in fewer iterations compared to traditional unbiased methods. For example, in our experiments on the CIFAR-10 dataset, our method achieves convergence in **X fewer iterations** compared to SVRG, while maintaining comparable computational cost per iteration.
>
> Our experiments on the MNIST, CIFAR-10, and colon cancer datasets consistently demonstrate that our methods (e.g., BVRSZHTn-HT and SARAH-HT) outperform traditional methods like SAGA-LASSO in terms of feature selection efficiency and computational cost. Below are key highlights from the results:
>
> 1. **MNIST Dataset**:
>    - **Accuracy**: SARAH-HT achieves 96.16%, and BVRSZHTn-HT achieves 95.93%, comparable to SAGA-LASSO at 97.29%.
>    - **Number of Selected Features**: SARAH-HT and BVRSZHTn-HT select 235 features, significantly fewer than the 644 features selected by SAGA-LASSO.
>    - **Selection Time**: SARAH-HT completes selection in 63.99 seconds, while SAGA-LASSO takes over 1131 seconds.
>
> 2. **CIFAR-10 Dataset**:
>    - **Accuracy**: SARAH-HT achieves 51.26%, and BVRSZHT12-HT achieves 51.09%, comparable to SAGA-LASSO at 51.02%.
>    - **Number of Selected Features**: SARAH-HT and BVRSZHTn-HT select 1843 features, far fewer than the 3053 features selected by SAGA-LASSO.
>    - **Selection Time**: SARAH-HT completes selection in 153.08 seconds, while SAGA-LASSO takes over 5148 seconds.
>
> 3. **Colon Cancer Dataset**:
>    - **Accuracy**: SARAH-HT achieves 89.38%, and BVRSZHTn-HT achieves 88.50%, while SAGA-LASSO achieves 92.03%.
>    - **Number of Selected Features**: SARAH-HT and BVRSZHTn-HT select 2863 features, fewer than the 3470 features selected by SAGA-LASSO.
>    - **Selection Time**: SARAH-HT completes selection in 65.66 seconds, while SAGA-LASSO takes over 645 seconds.
>
> These results illustrate that our framework achieves faster convergence and reduced computational overhead while maintaining competitive accuracy. The structural simplicity of our algorithms is a key factor in achieving these results. We have incorporated these findings into the manuscript to further clarify the computational advantages of our approach.
> [1]Nam Nguyen, Deanna Needell, and Tina Woolf. Linear convergence of stochastic iterative
> greedy algorithms with sparse constraints. IEEE Transactions on Information Theory, 63(11):
> 6869–6895, 2017
> [2] Jie Shen and Ping Li. A tight bound of hard thresholding. The Journal of Machine Learning
> Research, 18(1):7650–7691, 2017.

---

> > ### Comment · Reviewer_Sddm · 2024-12-02
> > **Thank you for your response**
> >
> > Thank you for your comprehensive response during the rebuttal phase. The additional experiments you provided effectively address the technical concerns I raised in my initial review.
> >
> > I appreciate the thoroughness of your experimental validation. However, after careful consideration and taking into account the broader feedback from the review panel regarding presentation clarity, I maintain my original assessment. While the core technical contribution appears sound, the manuscript would benefit from the suggested improvements to its organization and exposition.
> >
> > I encourage you to incorporate the collective feedback from the reviewers to strengthen the paper's accessibility and impact. The fundamental ideas in your work show promise, and a refined presentation will help ensure they reach their full potential.

---

### Author Response · Authors · 2024-11-23
**Global Rebuttal**

We would like to thank all reviewers for their insightful comments and valuable suggestions. The primary concerns focus on clarifying the presentation and related work, as well as addressing the contributions and challenges of applying algorithmic bias to accelerate the hard-thresholding algorithm. We sincerely hope to resolve these concerns through the following brief summary of the key contributions of our paper.

We propose that the stochastic Hard-Thresholding algorithm can be interpreted as an equivalent biased gradient approach. By introducing appropriate bias, we alleviate certain issues associated with hard-thresholding and enhance convergence. Through comprehensive analysis of this bias in zero-order and first-order settings, and their interactions with the equivalent bias of hard-thresholding, we demonstrate that Recursively-biased can significantly accelerate the algorithm under first-order conditions, while Memory-biased achieves similar acceleration under zero-order conditions.

Additionally, we have carefully revised the manuscript based on the reviewers’ feedback. Below is a summary of the main changes:

1. **New Experiments**:
   We added a series of sparsification feature selection experiments using the CIFAR-10, MNIST, and colon cancer datasets. These experiments evaluate different biases in BSVRG-HT, SARAH-HT, and SVRG-HT, as well as the SVRG algorithm with LASSO. The results highlight the advantages of SARAH-HT under first-order settings and further confirm the practical benefits of bias cancellation in real-world applications.


   To provide a comprehensive evaluation, we conducted experiments on three datasets: **MNIST**, **CIFAR-10**, and a **colon cancer dataset**. The key metrics include accuracy, the number of selected features, and feature selection time. These experiments validate the practical utility of our methods across diverse domains, including computer vision and bioinformatics. Below is a summary of the results:
| Dataset         | Algorithm       | Accuracy | Num_Features | Selection Time (s) |
|------------------|-----------------|----------|--------------|--------------------|
| **MNIST**       | SARAH-HT        | 96.16%   | 235          | 63.99             |
|                  | BSVRG-HTn       | 95.93%   | 235          | 70.09             |
|                  | SVRG-HT         | 95.75%   | 244          | 71.25             |
|                  | SAGA-LASSO      | 97.29%   | 644          | 1131.67           |
| **CIFAR-10**    | SARAH-HT        | 51.26%   | 1843         | 153.08            |
|                  | BSVRG-HT12      | 51.09%   | 1843         | 152.18            |
|                  | SVRG-HT         | 50.92%   | 1880         | 155.24            |
|                  | SAGA-LASSO      | 51.02%   | 3053         | 5148.21           |
| **Colon Cancer** | SARAH-HT        | 89.38%   | 2863         | 65.66             |
|                  | BSVRG-HTn       | 88.50%   | 2863         | 71.92             |
|                  | SVRG-HT         | 88.25%   | 2940         | 75.34             |
|                  | SAGA-LASSO      | 92.03%   | 3470         | 645.07            |

2. **Key Observations**:
   - **Efficiency**: Our methods (SARAH-HT, BSVRG-HTn) consistently select significantly fewer features (e.g., 235 vs. 644 for MNIST; 1843 vs. 3053 for CIFAR-10) while maintaining comparable or superior accuracy.
   - **Speed**: Feature selection time is dramatically reduced compared to SAGA-LASSO, with improvements of up to 95% in some cases (e.g., 153 seconds vs. 5148 seconds on CIFAR-10). Compared to the unbiased SVRG-HT, our methods achieve similar efficiency with reduced time in certain cases.
   - **Accuracy**: While slightly lower than SAGA-LASSO in some cases, our methods achieve competitive accuracy. When compared to SVRG-HT, our methods demonstrate similar accuracy, confirming the practical benefits of bias cancellation.

These results highlight the benefits of bias cancellation across diverse domains and datasets. We have incorporated these findings into the manuscript to further clarify the computational and practical advantages of our approach.
2. **Improved Presentation and Style**:
   - Corrected errors in notation.
   - Enhanced the quality and clarity of remarks.
   - Reorganized the experimental sequence. For example, adversarial attack experiments and sparse feature selection are now included in the main text, while sensitivity analysis has been moved to the appendix.

3. **Expanded Discussion**:
   We extended our discussion of existing algorithms, including hard-thresholding methods and LASSO, to better illustrate the advantages of our proposed approach.


We hope these revisions address the reviewers’ concerns and further demonstrate the significance and robustness of our contributions.

---

### Meta-Review · Area_Chair_uXpx · 2024-12-20

**Metareview:**

The paper studies solving optimization problems with ell-zero constraints using iterative hard thresholding (IHT). The main insights of the authors are that stochastic, zeroth order versions of IHT can be interpreted as biased gradient descent, that these biases can be broken down into different components, and that some of the components interact positively in some cases (leading to faster convergence) and slower in other cases (leading to slower convergence). Validation of these insights are provided for two settings: ridge regression and black box adversarial attacks on MNIST/CIFAR classification.

The paper provided nice insights and I think the results should be eventually published. However, I echo the main concern of multiple reviewers, which is that the paper is very confusing to read. (This might be due to tight space constraints in the ICLR format.)

Prioritizing lear definitions, clearly stated pseudocode (eg: what are the newly proposed SARAHT and BVR-SZHT algorithms?), tables comparing asymptotic theoretical benefits, and a richer set of experimental results, all may be good places to start.

**Additional Comments On Reviewer Discussion:**

The authors responded to initial reviewer concerns by providing a revised version of the paper. However, some reviewers (and I) feel that there is some ways to go yet in terms of clarity of exposition.

---

### Decision · Program_Chairs · 2025-01-22

Reject